# TrustworthyQENN: A Quantum Evidential Neural Network Based on Complex-Valued Contrastive Learning for Uncertainty Pattern Classification

Xiaolong Chen [1]  Fuyuan Xiao [1]  Xiaohong Zhang [1]  Zehong Cao [2]  Chin-Teng Lin [3]

## Abstract

Out-of-distribution (OOD) detection requires accurately classifying in-distribution (ID) samples while effectively distinguishing anomalous OOD data. However, existing methodologies predominantly rely on real-valued magnitude features, neglecting the semantic richness embedded in phase information, and often lack a systematic theoretical framework for quantitively modeling uncertainty. To address this dual limitation of incomplete feature representation and insufficient uncertainty modeling, the trustworthy quantum evidence neural network (TrustworthyQENN) is proposed, a novel quantum-inspired framework bridging complex-valued representation learning with generalized quantum evidence theory (GQET). Specifically, supervised complex-valued contrastive learning (SCVCL) is proposed to synchronize amplitude distributions with phase correlations, thereby enforcing high intra-class compactness and inter-class separability for ID data. A quantum evidence generation mechanism based on GQET is subsequently devised, where the OOD state is formally grounded as the quantum empty set within a Hilbert space. Furthermore, the generalized quantum evidential combination rule (GQECR) is leveraged to fuse multi-view evidence, thereby achieving trustworthy inference. Extensive experiments on the MSTAR, EuroSAT, and FUSAR-Ship benchmarks substantiate the superiority of TrustworthyQENN, which achieves a peak AUROC of 95.94% on the MSTAR dataset while consistently outperforming state-of-the-art methods across all evaluated scenarios.

[1]School of Big Data and Software Engineering, Chongqing University, Chongqing 401331, China [2]Adelaide University, Adelaide, SA, Australia [3]University of Technology Sydney, Sydney, NSW, Australia. Correspondence to: Fuyuan Xiao <doctorxiaofy@hotmail.com>.

*Proceedings of the 43$^{rd}$ International Conference on Machine Learning*, Seoul, South Korea. PMLR 306, 2026. Copyright 2026 by the author(s).

## 1. Introduction

Uncertainty is an omnipresent phenomenon that has emerged as a central research focus in information theory, dynamical and trustworthy machine learning (Wang et al., 2015b; 2018; Fan et al., 2025; Zhao et al., 2026). To effectively model and manage this uncertainty, scholars have proposed a diverse array of theoretical frameworks. Traditional approaches include Dempster-Shafer (D-S) evidence theory (Kang & Zhao, 2024; Yu et al., 2025) and belief rule-based systems (Yang & Xu, 2025a; Cao et al., 2023; Zhou et al., 2019), among others. More recently, novel developments have introduced random permutation sets (Deng, 2022; Deng et al., 2024a; Zhou et al., 2024b;a; Su et al., 2026a; Li et al., 2026), generalized divergence theory (Xiao et al., 2025a; Huang et al., 2023; 2024; Zhang & Xiao, 2024), entropy-based measures (Zhao et al., 2024; Zhan et al., 2024; Su et al., 2025), and extropy modeling (Li & Zhang, 2026; Zhan et al., 2026; Deng et al., 2024b). The proper handling of uncertainty is paramount across domains such as multi-source information fusion (Li et al., 2024; Xiahou et al., 2021; Liu et al., 2025; Guo et al., 2024; Meng et al., 2026), emergency decision-making (Fei et al., 2024; Song & Fei, 2026; Fei & Wang, 2022), risk assessment (Zhou et al., 2025; Chang et al., 2022; Chen & Deng, 2024), fault diagnosis (Zhang et al., 2026a), target recognition (Chenghai et al., 2025), and pattern classification (Zhang et al., 2022; 2026b; Liu et al., 2022; Zhang et al., 2024b; Sun et al., 2026), as it significantly enhances the robustness of network dynamics (Lei et al., 2025; 2026) and complex systems against unpredictable environments (Yang et al., 2024; Meng et al., 2023; Wang et al., 2022b; Li et al., 2018; Su et al., 2026b).

Among the multifaceted problems related to uncertainty, out-of-distribution (OOD) detection stands out as a critical challenge in reliable deployment. In practical applications, the testing environment inevitably contains samples that deviate from the training distribution, referred to as in-distribution (ID) data. These deviations manifest as semantic shifts (the appearance of unforeseen categories) or covariate shifts (changes in sensor conditions), introducing high levels of uncertainty (Wang et al., 2024; 2015a; 2017). While existing methods have achieved substantial progress

in distinguishing ID classes by leveraging diverse strategies, including structure-informed robust ensemble learning (Qiao et al., 2025), generalized statistical decision rules based on empirical p-values (Ma et al., 2025), progressive self-knowledge distillation (Yang & Xu, 2025b), and adversarial gradient attribution mechanisms (Zhang et al., 2025), they remain constrained by fundamental limitations when applied to complex-valued data domains: 1) *Incomplete data representation*. Predominant methodologies rely on real-valued magnitude features. This approach neglects the semantic richness embedded in phase information, a critical component that is essential for capturing the physical geometry of signals like SAR (Shi et al., 2025; Xiao et al., 2024). 2) *Coarse-grained uncertainty modeling*. Conventional approaches (Xiao et al., 2025c) often lack a systematic theoretical framework to explicitly model the epistemic uncertainty arising from OOD samples; instead, most methods merely use heuristic thresholds to distinguish OOD samples rather than formalizing the OOD state within the theoretical framework.

During uncertainty processing, evidence theory has proven to be an effective tool (Han et al., 2018; Deng et al., 2023; Du et al., 2025; Zhang et al., 2024a). However, classical approaches exhibit deficiencies when handling OOD problems and complex-valued data. In contrast, generalized quantum evidence theory (GQET) (Xiao, 2023; Aug. 2025), proposed by Xiao, offers distinct advantages by addressing uncertainty within a Hilbert space in the open world. By employing generalized quantum basic probability amplitudes (GQBPAs), GQET characterizes fine-grained uncertainty (Yu et al., 2026; Xiao et al., 2025b). Furthermore, the generalized quantum evidential combination rule (GQECR) leverages quantum constructive interference to continuously and dynamically fuse multi-dimensional features. Ultimately, by integrating the fine-grained representation of GQBPAs, the dynamic fusion of GQECR, and the establishment of trustworthy confidence intervals, GQET provides a superior, unified mechanism to explicitly model the uncertainty arising from OOD classes.

To address the aforementioned dual limitations of representation and uncertainty modeling, the trustworthy quantum evidence neural network (TrustworthyQENN) is proposed, a novel quantum-inspired framework bridging complex-valued representation learning with generalized quantum evidential reasoning. First, to ensure data representation completeness, a supervised complex-valued contrastive learning (SCVCL) mechanism is proposed. This mechanism synchronizes amplitude distributions with phase correlations, learning a feature where ID data exhibits high compactness, thereby naturally isolating OOD regions. Second, to achieve fine-grained uncertainty modeling, the OOD state is not merely treated as a rejection threshold, but formally grounded as the quantum empty set within a Hilbert space.

By mapping complex-valued features to the generalized quantum basic probability amplitude (GQBPA) function, the generalized quantum evidential combination rule (GQECR) is leveraged to fuse multi-view evidence, thereby achieving trustworthy inference. Our specific contributions are as follows:

- Supervised complex-valued contrastive learning (SCVCL) is proposed, which enforces intra-class compactness and inter-class separability in the complex-valued domain. By jointly optimizing amplitude and establishing linear separability in phase-aware embeddings, SCVCL captures more complete semantic information, overcoming the representation bottleneck of real-valued baselines for ID classification.

- A quantum evidence generation mechanism is designed to bridge the gap between feature representations and evidential reasoning. This mechanism explicitly models OOD samples as the empty set within the quantum-inspired framework, enabling precise quantification of uncertainty for anomaly detection.

- A Quantum Evidential Fusion strategy is implemented using the GQECR. This approach leverages quantum-mechanical properties to resolve conflicts from multi-view evidence, achieving state-of-the-art performance on the MSTAR, EuroSAT, and FUSAR-Ship benchmarks under both semantic and domain shift scenarios.

The remainder of this paper is organized as follows: Section 2 details the key modules of the proposed method; Section 3 presents the experimental implementation and results across multiple datasets; finally, Section 4 concludes the paper.

## 2. Quantum Evidence Generation Neural Network

### 2.1. Problem Statement

The core challenge of OOD detection is to accurately classify ID samples while effectively detecting OOD samples during inference. Formally, let $\mathcal{X} \subset \mathbb{R}^{\mathsf{C} \times \mathsf{H} \times \mathsf{W}}$ denote the input space, where $\mathsf{C}, \mathsf{H}, \mathsf{W}$ represent the number of channels, height, and width, respectively. Let $\mathcal{D}_{\mathrm{in}} = \{(x_t, y_t)\}_{t=1}^{\mathcal{N}}$ denote the ID training set, where $x_t \in \mathcal{X}$ represents the input observation and $y_t \in \mathcal{Y}_{in}$ is the corresponding label from the ID label set $\mathcal{Y}_{in} = \{1, \ldots, K\}$. The objective is to train a model $\mathcal{F} : \mathcal{X} \to \mathbb{C}^K$ that minimizes classification error on $\mathcal{Y}_{in}$ while accurately identifying test samples $\{(x_v, y_v)\}_{v=1}^{\mathcal{M}}$ where $y_v \notin \mathcal{Y}_{in}$ as OOD data.

The SCVCL comprises three modules: the complex-valued encoding network $E(\cdot)$, the complex-valued projection network $G(\cdot)$, and the complex-valued classification network

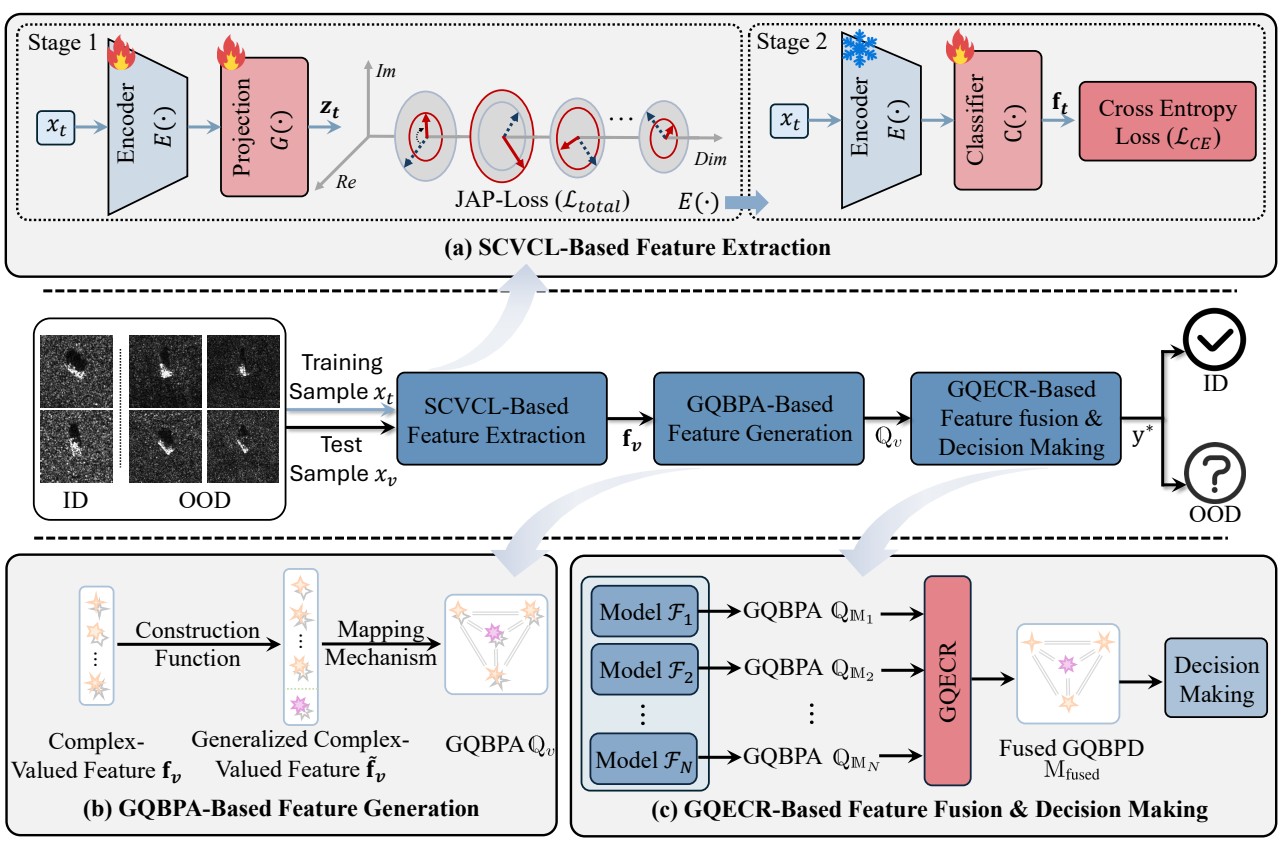

*Figure 1.* Overall framework diagram of TrustworthyQENN.

$C(\cdot)$. All modules are instantiated as complex-valued neural networks (detailed parameters are provided in Section 3. A two-stage training paradigm is employed to collaboratively optimize representation learning and classification performance. In the first stage, $E(\cdot)$ and $G(\cdot)$ operate cooperatively to map the input $x_t$ to a complex-valued projection vector $\mathbf{z}_t = G(E(x_t))$. Guided by the proposed joint amplitude-phase supervision, $\mathbf{z}_t$ learns to synchronize amplitude distributions with phase correlations. In the second stage, the parameters of $E(\cdot)$ are frozen, and $C(\cdot)$ is fine-tuned via the cross-entropy loss to yield discriminative complex-valued feature $\mathbf{f}_t = C(E(x_t))$.

### 2.2. Method Overview

To overcome the limitations of conventional real-valued OOD detection models to fully utilize the phase information of complex-valued data, and to resolve the difficulties in generating GQBPA function for GQET, a novel framework, termed the trustworthy quantum evidence neural network (TrustworthyQENN), is proposed. As shown in Fig. 1, TrustworthyQENN establishes a pipeline that bridges raw data, quantum evidential representations, and final decisions. The architecture comprises three integral components. First, su-

pervised complex-valued contrastive learning for extracting discriminative complex-valued features. Second, a novel mapping mechanism that transforms complex-valued features into valid quantum evidence. Finally, a quantum evidence fusion module leverages the GQECR to integrate evidence and reach dependable decisions.

### 2.3. Supervised Complex-Valued Contrastive Learning-Based Feature Extraction

Contrastive learning optimizes the feature space by constructing positive and negative sample pairs, a process recognized for enhancing intra-class compactness and inter-class separability. This paradigm aligns intrinsically with the core objective of OOD detection, which is to maximize the margin between ID and OOD distributions. However, existing contrastive learning frameworks are predominantly defined in the real-valued domain, focusing solely on feature amplitude while neglecting the unique semantic information contained in the phase of complex-valued data. Consequently, a novel supervised complex-valued contrastive learning (SCVCL) method is proposed, enabling the joint learning of amplitude and phase information while preserving the fundamental advantages of contrastive learning.

**Definition 2.1** (Amplitude Loss). Let $\mathcal{B}$ denote the set of indices for a batch of complex-valued projection vectors. For an anchor index $t \in \mathcal{B}$ with class label $y_t$, let $\mathbf{a}_t \in \mathbb{R}^D$ represent the amplitude component derived from $D$-dimensional anchor complex-valued projection vector $\mathbf{z}_t$, defined as $\mathbf{a}_t = |\mathbf{z}_t|$. Furthermore, let $\Omega(t) = \{j \in \mathcal{B} \setminus \{t\} \mid y_j = y_t\}$ denote the set of indices for positive samples sharing the same label. To enforce discriminability within the amplitude domain, the amplitude loss is formulated to maximize the cosine similarity between intra-class amplitude vectors. The amplitude loss is defined as:

$$\mathcal{L}_{amp} = \sum_{t \in \mathcal{B}} \frac{-1}{|\Omega(t)|} \sum_{j \in \Omega(t)} \log \frac{\exp\left(\text{sim}(\mathbf{a}_t, \mathbf{a}_j)/\tau\right)}{\sum_{\varsigma \in \mathcal{B} \setminus \{t\}} \exp\left(\text{sim}(\mathbf{a}_t, \mathbf{a}_\varsigma)/\tau\right)}, \quad (1)$$

where $\tau$ is a scalar temperature parameter, and $\text{sim}(\cdot, \cdot)$ denotes the cosine similarity operator, which is employed to optimization of distance relationships between complex-valued projection vectors.

**Definition 2.2** (Phase Projection Vector). Let $\boldsymbol{\theta}_t = [\theta_t^{(1)}, \cdots, \theta_t^{(k)}, \cdots, \theta_t^{(D)}]$ be the raw phase vector extracted from $\mathbf{z}_t$, where $\theta_t^{(k)} \in [-\pi, \pi]$. Directly applying standard norms to $\boldsymbol{\theta}_t$ is problematic, as the numerical distance between $-\pi$ and $\pi$ is maximized despite these angles being geometrically identical. Inspired by (Yang & Yan, 2022), to resolve this cyclic ambiguity and map the phase data into a continuous Euclidean space, a *phase projection vector* $\mathbf{p}_t \in \mathbb{R}^{2D}$ is constructed by mapping each angular component to the Cartesian coordinate system:

$$\mathbf{p}_t = \left[\cos\theta_t^{(1)}, \sin\theta_t^{(1)}, \ldots, \cos\theta_t^{(D)}, \sin\theta_t^{(D)}\right]^\top. \quad (2)$$

A critical property of this transformation is that the Euclidean-based cosine similarity between two projection vectors, $\mathbf{p}_t$ and $\mathbf{p}_j$, is mathematically equivalent to the mean cosine of their angular differences:

$$\text{sim}(\mathbf{p}_t, \mathbf{p}_j) = \frac{\mathbf{p}_t^\top \mathbf{p}_j}{\|\mathbf{p}_t\|_2 \|\mathbf{p}_j\|_2} = \frac{1}{D} \sum_{k=1}^{D} \cos(\theta_t^{(k)} - \theta_j^{(k)}). \quad (3)$$

Detailed derivations and analyses of the *phase projection vector* are provided in **Appendix B**. This transformation linearizes the phase space, enabling the effective application of standard Euclidean-based contrastive objectives.

**Definition 2.3** (Phase Loss). Let $\Omega(t)$ denote the set of indices for positive samples corresponding to the anchor $t$. To align the semantics encoded in the phase information, the phase loss is formulated to maximize the consistency between phase projection vectors of the same class. The phase loss is defined as:

$$\mathcal{L}_{pha} = \sum_{t \in \mathcal{B}} \frac{-1}{|\Omega(t)|} \sum_{j \in \Omega(t)} \log \frac{\exp\left(\text{sim}(\mathbf{p}_t, \mathbf{p}_j)/\tau\right)}{\sum_{\varsigma \in \mathcal{B} \setminus \{t\}} \exp\left(\text{sim}(\mathbf{p}_t, \mathbf{p}_\varsigma)/\tau\right)}. \quad (4)$$

**Definition 2.4** (Joint Amplitude-Phase Loss, JAP-Loss). Let $\alpha \in [0, 1]$ be a balancing hyperparameter. To simultaneously optimize the amplitude and phase components, the JAP-Loss is defined as a weighted summation:

$$\mathcal{L}_{total} = (1 - \alpha)\mathcal{L}_{amp} + \alpha\mathcal{L}_{pha}. \quad (5)$$

After contrastive learning, the encoder $E(\cdot)$ is frozen and the classifier $C(\cdot)$ is optimized to produce class-discriminative feature $\mathbf{f}_t \in \mathbb{C}^{|\mathcal{Y}_{in}|}$. The cross-entropy loss is applied to the feature modulus $|\mathbf{f}_t|$, bridging the complex-valued domain with categorical labels:

$$\mathcal{L}_{CE} = -\sum_{k=1}^{|\mathcal{Y}_{in}|} \mathbb{I}(y_t = k) \log \left( \frac{\exp(|\mathbf{f}_t^{(k)}|)}{\sum_{j=1}^{|\mathcal{Y}_{in}|} \exp(|\mathbf{f}_t^{(j)}|)} \right), \quad (6)$$

where $\mathbb{I}(\cdot)$ is the indicator function. This ensures feature magnitudes reflect classification confidence, preparing them for evidential mapping.

## 2.4. GQBPA-Based Feature Generation

For a test sample $x_v$, although the complex-valued feature $\mathbf{f}_v = C(E(x_v)) \in \mathbb{C}^{|\mathcal{Y}_{in}|}$ derived from SCVCL exhibits discriminative capability, it lacks the inherent capacity to model uncertainty. Therefore, it is necessary to construct a generalized quantum basic probability amplitude (GQBPA) that allows for the explicit expression of uncertainty, utilizing the empty set $\emptyset$ to represent the OOD. To bridge the gap between deterministic feature vectors and evidential reasoning, a novel complex-valued feature to quantum evidence mapping mechanism is established. This process involves the construction of a generalized complex-valued feature vector augmented with an uncertainty dimension, followed by a normalization mapping procedure to generate a valid GQBPA.

### 2.4.1. GENERALIZED COMPLEX-VALUED FEATURE VECTOR

The construction of the generalized complex-valued feature vector $\tilde{\mathbf{f}}_v$ is predicated on the statistical analysis of feature distributions. The core objective is to quantify the "ID-ness" of a sample and allocate the residual probability to the OOD state.

**Definition 2.5** (Mean Complex-Valued Activation Vector, MCVAV). Let $S_k$ denote the set of indices of training samples correctly classified as class $k \in \mathcal{Y}_{in}$. The prototypical representation for each ID class $k$ is established by computing the centroid of the complex-valued features corresponding to these indices. The MCVAV $\boldsymbol{\mu}_k$ is defined as:

$$\boldsymbol{\mu}_k = \frac{1}{|S_k|} \sum_{t \in S_k} \mathbf{f}_t. \quad (7)$$

$\boldsymbol{\mu}_k$ will serve as the class center for class $k$, providing an evaluation basis for the attribution of subsequent samples.

**Definition 2.6** (Generalized Complex-Valued Feature Reliability Weight). Let $d(\mathbf{f}_v, \boldsymbol{\mu}_k) = \|\mathbf{f}_v - \boldsymbol{\mu}_k\|_2$ define the distance metric between a sample and the class prototype. According to (Bendale & Boult, 2016), the distances typically

follow a Weibull distribution at the tails. Let $(\sigma_k, \gamma_k, \beta_k)$ be the Weibull parameters estimated from the training data for class $k$, where $\sigma_k$, $\gamma_k$, and $\beta_k$ denote the scale, shape, and location parameters, respectively. The reliability weight $w_k(\mathbf{f}_v)$, representing the probability that $\mathbf{f}_t$ is not an outlier. The $w_k(\mathbf{f}_v)$ is defined as:

$$w_k(\mathbf{f}_v) = \exp\left(-\left(\frac{d(\mathbf{f}_v, \boldsymbol{\mu}_k) - \beta_k}{\sigma_k}\right)^{\gamma_k}\right). \qquad (8)$$

A value of $w_k(\mathbf{f}_v)$ approaching unity indicates strong adherence to the ID distribution, whereas a value near 0 signals potential OOD characteristics.

**Definition 2.7** (Generalized Complex-Valued Feature Construction Function). Let the feature space be explicitly augmented from dimension $|\mathcal{Y}_{in}|$ to $|\mathcal{Y}_{in}| + 1$ to accommodate the OOD state. The augmented quantum activation vector $\tilde{\mathbf{f}}_v \in \mathbb{C}^{|\mathcal{Y}_{in}|+1}$ is synthesized by recalibrating known activations and aggregating uncertainty:

$$\tilde{\mathbf{f}}_v = [\tilde{f}_v^{(1)}, \ldots, \tilde{f}_v^{(|\mathcal{Y}_{in}|)}, \tilde{f}_v^{(|\mathcal{Y}_{in}|+1)}]^\top. \qquad (9)$$

For ID classes $k \in \mathcal{Y}_{in}$, the activation is modulated by the reliability weight to penalize low-confidence predictions:

$$\tilde{f}_v^{(k)} = f_v^{(k)} \cdot w_k(\mathbf{f}_v). \qquad (10)$$

The activation for the OOD dimension (indexed as $|\mathcal{Y}_{in}|+1$) is synthesized by accumulating the "discounted" confidence from all ID classes:

$$\tilde{f}_v^{(|\mathcal{Y}_{in}|+1)} = \sum_{k=1}^{|\mathcal{Y}_{in}|} f_v^{(k)}(1 - w_k(\mathbf{f}_v)). \qquad (11)$$

This formulation ensures that if a sample acts as an outlier across all ID class distributions (i.e., $\forall k, w_k \to 0$), the amplitude of the OOD component $\tilde{f}_v^{(|\mathcal{Y}_{in}|+1)}$ becomes dominant, thereby facilitating the detection of novel inputs.

### 2.4.2. GENERALIZED QUANTUM BASIC PROBABILITY AMPLITUDE MAPPING MECHANISM

To formally ground the network outputs within GQET, the quantum frame of discernment (QFOD) is first established. Let $\Phi = \{\phi_1, \ldots, \phi_{|\mathcal{Y}_{in}|}\}$ represent the set of ID classes, where each $\phi_k$ is an orthogonal basis vector in the Hilbert space. The power set $2^\Phi$ encompasses all possible singleton hypotheses corresponding to specific classes, as well as the empty set $\emptyset$ which explicitly represents the OOD state. The augmented vector $\tilde{\mathbf{f}}_v \in \mathbb{C}^{|\mathcal{Y}_{in}|+1}$ derived from the previous stage serves as the precursor to probability assignment. However, $\tilde{\mathbf{f}}_v$ necessitates further constraints to strictly adhere to the normalization axioms of GQET.

**Definition 2.8** (Generalized Quantum Basic Probability Amplitude Mapping Function). Let $\xi(\tilde{\mathbf{f}}_v) = e^{|\tilde{\mathbf{f}}_v|}/|\tilde{\mathbf{f}}_v|$ be defined as the quantum probability amplification factor (detailed derivations and theoretical analyses are provided in **Appendix C**) to enhance evidential contrast (inspired by

the Softmax mechanism's ability to amplify feature disparities). The mapping from the complex-valued feature space to the GQBPA, denoted as $\mathbb{Q}_v : 2^\Phi \to \mathbb{C}$, is constructed via normalization:

$$\mathbb{Q}_v(\psi_j) = \begin{cases} \frac{\tilde{\mathbf{f}}_v^{(j)} \cdot \xi(\tilde{\mathbf{f}}_v^{(j)})}{\aleph_v}, & \text{if } \psi_j = \phi_j \in \Phi \text{ or } \psi_j = \emptyset, \\ 0, & \text{otherwise,} \end{cases} \qquad (12)$$

where $\aleph_v = \sqrt{\sum_{\epsilon=1}^{|\mathcal{Y}_{in}|+1} \left|\tilde{\mathbf{f}}_v^{(\epsilon)} \cdot \xi(\tilde{\mathbf{f}}_v^{(\epsilon)})\right|^2}$ is the normalization factor.

This formulation guarantees that $\mathbb{Q}_v$ strictly adheres to the mathematical constraints of GQET. Specifically, the summation of the squared magnitudes of the GQBPA is rigorously shown to equal unity:

$$\sum_{\psi \in 2^\Phi} |\mathbb{Q}_v(\psi)|^2 = \sum_{j=1}^{|\mathcal{Y}_{in}|+1} \frac{\left|\tilde{\mathbf{f}}_v^{(j)} \cdot \xi(\tilde{\mathbf{f}}_v^{(j)})\right|^2}{\aleph_v^2} = 1. \qquad (13)$$

Consequently, $\mathbb{Q}_v$ constitutes a valid GQBPA, ready for subsequent fusion operations.

### 2.5. GQECR-Based Information Fusion and Decision Making

Inference derived from a singular source is inherently susceptible to data fluctuations and inherent variability. To enhance predictive robustness, an ensemble strategy is adopted, deploying $N$ independent TrustworthyQENN models initialized with different random seeds to ensure diversity. In contrast to classical Dempster-Shafer theory (DST), which primarily relies on the algebraic intersection of probability masses to combine evidence, GQET characterizes the interaction between evidence views by incorporating quantum interference effects.

The fusion of multi-source evidence is governed by the GQECR. Let $\{\mathbb{Q}_{\mathbb{M}_1}, \mathbb{Q}_{\mathbb{M}_2}, \ldots, \mathbb{Q}_{\mathbb{M}_N}\}$ represent the set of independent GQBPAs generated by the ensemble. By applying GQECR (Definition A.8), all GQBPAs are fused to obtain a generalized quantum basic probability distribution (GQBPD) $\mathbb{M}_{\text{fused}}$

$$\mathbb{M}_{\text{fused}} = \mathbb{Q}_{\mathbb{M}_1} \oplus \mathbb{Q}_{\mathbb{M}_2} \oplus \cdots \oplus \mathbb{Q}_{\mathbb{M}_N}. \qquad (14)$$

This operation effectively integrates the amplitude and phase information from multiple views, yielding a more comprehensive assessment of the target identity.

The final decision logic is predicated on the support degree $\mathbb{M}(\psi)$. The predicted label $y^*$ is determined as follows:

$$y^* = \begin{cases} \text{OOD}, & \text{if } \arg\max_{\psi_j \in \Phi} \mathbb{M}(\psi_j) = \emptyset \\ & \text{or } \max \mathbb{M}(\cdot) < \delta, \\ \arg\max_{k \in \mathcal{Y}_{kn}} \mathbb{M}(\phi_k), & \text{otherwise,} \end{cases} \qquad (15)$$

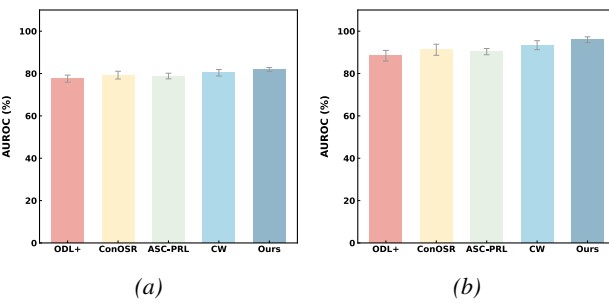

*Figure 2.* Performance comparison of AUROC metrics with (a) ID: MSTAR / OOD: FUSAR-ship and (b) ID: FUSAR-ship / OOD: MSTAR.

where $\delta$ is a pre-calibrated probability threshold. A sample is rejected as OOD if the evidence explicitly supports the empty set $\emptyset$, or if the maximum support for any ID class is insufficient to meet the confidence standard $\delta$. This mechanism ensures that the system maintains high precision on ID classes while exhibiting robust rejection capabilities for out-of-distribution inputs.

## 3. Experiments

This section validates TrustworthyQENN across the MSTAR, EuroSAT, and FUSAR-Ship datasets. After outlining the implementation details, we present a comparative analysis against leading baselines. We conclude with extensive ablation and sensitivity studies to elucidate the effectiveness of the proposed complex-valued and quantum evidential mechanisms.

### 3.1. Datasets and Experimental Setup

To ensure a fair comparison, all methods are evaluated on three standard benchmarks: MSTAR (Xiao et al., 2025c), EuroSAT (Helber et al., 2019), and FUSAR-Ship (Wang et al., 2022a). These datasets cover various modalities, including SAR and multi-spectral imagery. For each dataset, classes are partitioned into ID classes ($L_{in}$) and OOD classes ($L_{out}$). Detailed dataset statistics, class partitions, and the definition of openness ($\mathbb{O}$) are provided in **Appendix** D. The TrustworthyQENN is implemented based on a complex-valued convolutional architecture. The training process employs a two-stage paradigm using the Adam optimizer. Following (Liu et al., 2022; Bendale & Boult, 2016), the four standard metrics are adopted for evaluation: AUROC, OSCR, FPR95, and ACC. Comprehensive details regarding the network architecture, hyperparameter settings (e.g., learning rate, temperature $\tau$, ensemble size), and evaluation protocols are specified in **Appendix** D.

### 3.2. Comparative Experiments

The quantitative results are summarized in Table 1, comparing the proposed TrustworthyQENN against baseline methods and recent state-of-the-art OOD methods. To ensure a comprehensive evaluation, the selected comparison methods cover a wide range of technical paradigms: Softmax, OpenMax(Bendale & Boult, 2016), ODL / ODL+(Liu et al., 2022), ARPL(Chen et al., 2021), ConOSR(Xu et al., 2023), ASC-RPL(Xiao et al., 2025c), DEF(Li et al., 2025), and CW(Wang et al., 2025).

As observed, TrustworthyQENN achieves superior performance across all metrics. As summarized in Table 1, TrustworthyQENN demonstrates an advantage on the MSTAR dataset, achieving an AUROC of 95.94% and reducing FPR95 to 53.14%, which underscores the model's ability to utilize phase-amplitude coupling through the SCVCL mechanism for superior anomaly detection in the complex-valued domain. This performance lead consistently extends to the EuroSAT and FUSAR-Ship datasets, where TrustworthyQENN outperforms CW by 3.58% on EuroSAT and 2.17% on FUSAR-Ship in terms of AUROC. Compared to contemporary uncertainty-centric approaches like CW, TrustworthyQENN exhibits enhanced stability and theoretical robustness by explicitly modeling the OOD as the quantum empty set $\emptyset$ within a Hilbert space, with observed lower standard deviations ($\pm 1.5\%$ on MSTAR) confirming that the synergy between complex-valued representations and quantum evidential fusion yields a highly reliable framework for OOD.

A pervasive challenge in OOD is the "accuracy-rejection trade-off", where aggressive rejection strategies often compromise the classification accuracy of ID classes. To evaluate this, we report the ACC in Table 2. It is observed that TrustworthyQENN does not sacrifice closed-set performance; conversely, it achieves the highest accuracy across all datasets. On MSTAR, TrustworthyQENN reaches 99.48%, and on the challenging FUSAR-Ship, it achieves 96.79%, outperforming the second-best method. This phenomenon can be attributed to the proposed SCVCL. By synchronizing amplitude distributions with phase correlations, SCVCL enforces high intra-class compactness within the complex-valued domain. Consequently, the ID class prototypes become highly distinct, which not only facilitates the rejection of anomalies (as seen in Table 1) but also enhances the granularity of classification within the ID set.

### 3.3. Ablation Experiments

To validate the contributions of the proposed components, we perform ablation studies as detailed in Table 3. The results indicate that the *Complex-Valued Baseline* significantly outperforms its real-valued counterpart (e.g., a 4.00% AUROC gain on MSTAR), confirming that phase information

*Table 1.* OOD detection results in terms of the AUROC(%), OSCR(%) and FPR95(%). The best method is emphasized in **bold**, and the underlined denotes the second best result.

| Method | MSTAR | | | EuroSAT | | | FUSAR-Ship | | |
|---|---|---|---|---|---|---|---|---|---|
| | AUROC↑ | OSCR↑ | FPR95↓ | AUROC↑ | OSCR↑ | FPR95↓ | AUROC↑ | OSCR↑ | FPR95↓ |
| Softmax | 85.13±1.2 | 83.45±2.1 | 63.20±0.9 | 76.82±1.8 | 73.11±2.5 | 70.05±1.5 | 71.68±2.3 | 70.30±1.1 | 73.54±1.8 |
| OpenMax | 85.56±2.0 | 83.22±1.5 | 62.18±1.3 | 77.45±2.2 | 74.89±0.9 | 68.33±2.6 | 72.12±1.7 | 69.03±2.4 | 74.35±1.0 |
| ODL | 85.79±1.7 | 83.55±2.3 | 61.24±0.7 | 78.20±1.4 | 75.67±2.0 | 67.50±1.2 | 72.85±2.7 | 69.53±1.6 | 73.67±2.1 |
| ODL+ | 85.92±2.5 | 83.94±1.1 | 60.13±1.8 | 79.50±0.8 | 77.28±1.9 | 66.23±2.2 | 73.54±1.3 | 70.17±0.9 | 72.88±2.8 |
| ARPL | 86.21±1.0 | 84.52±2.7 | 57.58±2.4 | 81.52±2.1 | 79.85±1.6 | 62.01±0.9 | 74.85±2.0 | 70.89±2.3 | 71.17±1.4 |
| ConOSR | 86.47±2.2 | 84.86±1.8 | 58.74±1.5 | 83.00±2.6 | 81.20±0.7 | 61.39±1.1 | 75.50±2.4 | 71.17±1.9 | 68.97±2.0 |
| ASC-RPL | 86.45±1.4 | 85.30±2.4 | 56.20±2.1 | 81.33±1.2 | 77.52±2.3 | 63.17±1.7 | 74.91±0.8 | 73.22±2.6 | 71.13±1.8 |
| DEF | 90.15±2.8 | 89.30±1.3 | 55.45±1.0 | 84.53±1.9 | 83.21±2.5 | 60.67±2.0 | 78.71±1.6 | 76.38±0.7 | 68.45±2.2 |
| CW | 95.01±1.6 | 94.16±2.0 | 54.41±1.2 | 85.95±2.3 | 84.48±1.7 | 59.03±2.7 | 80.50±1.1 | 76.60±2.4 | 67.46±2.9 |
| **Ours** | **95.94±1.5** | **95.93±1.9** | **53.14±1.4** | **89.53±1.0** | **89.50±1.2** | **57.14±1.8** | **82.67±1.1** | **82.19±0.8** | **66.23±2.6** |

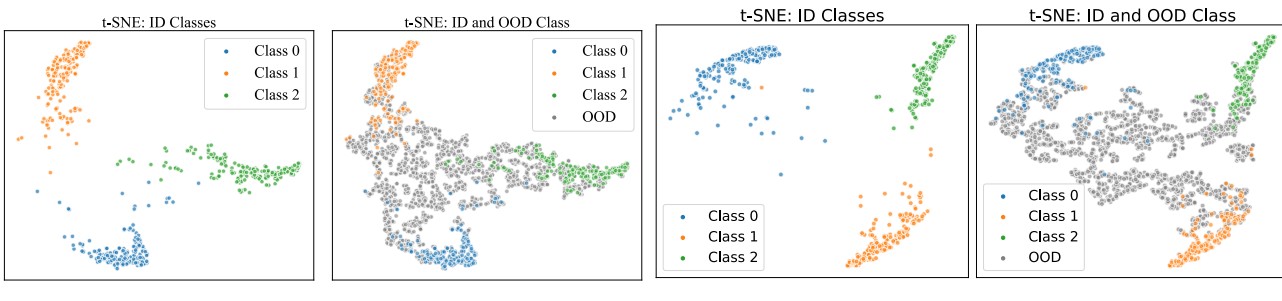

*(a)* Projection Features  *(b)* Classifier Features

*Figure 3.* t-SNE visualization of feature distributions on the MSTAR dataset. (a) Visualization of Projection Vectors, illustrating the initial clustering. (b) Visualization of Classifier Features, demonstrating enhanced compactness and clear separation between ID classes and OOD samples.

*Table 2.* Comparisons of Closed Set Accuracy (ACC %) on three datasets. The best method is emphasized in **bold**.

| Method | MSTAR | EuroSAT | FUSAR |
|---|---|---|---|
| Softmax | 97.09 | 92.48 | 89.79 |
| Openmax | 97.23 | 92.96 | 90.60 |
| ConOSR | 98.87 | 95.48 | 92.05 |
| ASC-RPL | 98.83 | 95.40 | 91.88 |
| DEF | 99.09 | 95.93 | 92.74 |
| CW | 99.12 | 96.31 | 93.01 |
| TrustworthyQENN | **99.48** | **98.37** | **96.79** |

captures critical structural semantics essential for distinguishing ID from OOD patterns. Furthermore, comparing the *w/o QECR* variant with the full TrustworthyQENN demonstrates that the GQECR provides a 3.31% AUROC improvement on FUSAR-Ship, primarily due to its ability to explicitly model the quantum empty set ∅ and leverage quantum-mechanical interference to resolve uncertainty. Finally, the removal of JAP-Loss leads to a marked performance decline (e.g., OSCR drops to 93.47% on MSTAR), proving that joint amplitude-phase optimization is indispensable for constructing a compact and discriminative feature space that effectively bounds the open space.

### 3.4. Cross-Dataset Analysis

To rigorously evaluate the robustness of TrustworthyQENN against significant domain shifts, we conducted a reciprocal "cross-dataset" experiment between MSTAR and FUSAR-Ship. These datasets originate from different sensors with distinct imaging parameters, creating a substantial domain gap. As depicted in Fig. 2, we present the AUROC performance for two reciprocal scenarios. In the scenario where MSTAR serves as the ID dataset (Fig. 2(a)), TrustworthyQENN demonstrates strong generalization, effectively distinguishing the unseen FUSAR-Ship targets. More notably, in the reverse setting where FUSAR-Ship is the ID dataset (Fig. 2(b)), TrustworthyQENN maintains a consistent performance advantage, significantly outperforming the runner-up method CW. Detailed comparisons for OSCR and FPR95 metrics are provided in **Appendix** E.1.

### 3.5. Feature Visualization

To intuitively validate the discriminative capability of the learned representations, we employ t-Distributed Stochastic Neighbor Embedding (t-SNE) to visualize the feature distributions on the MSTAR dataset. Given that standard

*Table 3.* Ablation experimental results in terms of the AUROC(%), OSCR(%) and FPR95(%). The best method is emphasized in **bold**, and the underlined denotes the second best result.

| Method | MSTAR | | | EuroSAT | | | FUSAR-Ship | | |
|---|---|---|---|---|---|---|---|---|---|
| | AUROC↑ | OSCR↑ | FPR95↓ | AUROC↑ | OSCR↑ | FPR95↓ | AUROC↑ | OSCR↑ | FPR95↓ |
| Real-Valued Baseline | 86.47 | 84.86 | 58.74 | 83.00 | 81.20 | 61.39 | 75.50 | 71.17 | 68.97 |
| Complex-Valued Baseline | 90.47 | 90.39 | 57.29 | 83.85 | 83.81 | 60.31 | 78.01 | 77.49 | 70.52 |
| w/o GQECR | 91.24 | 91.19 | 56.60 | 84.16 | 84.13 | 59.58 | 79.36 | 78.85 | 69.86 |
| w/o JAP-Loss | 93.47 | 93.47 | 55.03 | 88.66 | 88.62 | 58.49 | 82.09 | 81.60 | 69.31 |
| **TrustworthyQENN** | **95.94** | **95.93** | **53.14** | **89.53** | **89.50** | **57.14** | **82.67** | **82.19** | **66.23** |

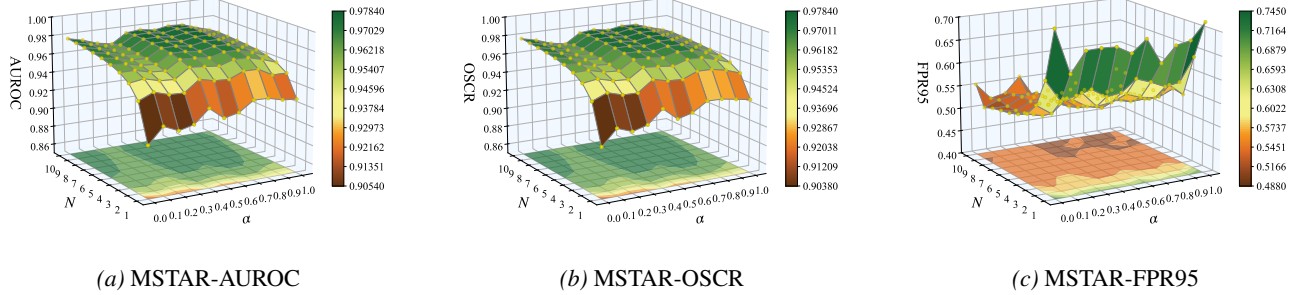

*(a)* MSTAR-AUROC          *(b)* MSTAR-OSCR          *(c)* MSTAR-FPR95

*Figure 4.* Sensitivity analysis of the TrustworthyQENN on the MSTAR dataset with respect to the balance parameter $\alpha$ and ensemble size $N$.

t-SNE is defined in the real domain, we adapt it for our complex-valued features $\mathbf{z} \in \mathbb{C}^D$ by concatenating the real and imaginary components into a $2D$-dimensional real-valued vector $[\Re(\mathbf{z}), \Im(\mathbf{z})]$ prior to dimensionality reduction. Fig. 3 illustrates the evolution of the feature vectors from the projection space to the final classification space. Fig. 3(a) depicts the distribution of the projection vectors derived from the encoder and projection head. The left panel of Fig. 3(a) (ID Only) demonstrates that the SCVCL mechanism successfully clusters the three ID classes into distinct regions. However, as shown in the right panel of Fig. 3(a) (ID + OOD), when OOD samples are introduced, significant overlap persists between the peripheral distributions of ID classes and the OOD samples. This indicates that while the projection space captures general semantic structures, the boundaries for OOD detection are not yet fully solidified. In contrast, Fig. 3(b) visualizes the classifier features obtained after the fine-tuning stage. Compared to the projection vectors, these features exhibit substantially enhanced intra-class compactness and inter-class separability. Notably, the decision boundaries are sharper, and the OOD samples are effectively pushed to the low-density regions of the feature space, forming a clear separation from the ID class prototypes.

### 3.6. Sensitivity Analysis

To assess the robustness of TrustworthyQENN, we evaluate the impact of the balance parameter $\alpha$ and ensemble size $N$

on the MSTAR dataset (Fig. 4) (the experimental results of other datasets are presented in the **Appendix** E.2). Results show that optimal performance is achieved at $\alpha = 0.4$, indicating that a balanced weighting of amplitude and phase information yields the most discriminative feature for SAR ground target recognition. Furthermore, performance exhibits a consistent positive correlation with the ensemble size $N \in [1, 10]$; as $N$ increases, the QECR effectively aggregates diverse model evidence to suppress conflicting noise and sharpen decision boundaries, leading to steady improvements in AUROC and OSCR scores alongside a reduction in FPR95.

## 4. Conclusion

This paper presents TrustworthyQENN, a novel framework that unifies complex-valued representation learning with generalized quantum evidence theory to address the critical challenge of out-of-distribution (OOD) detection. Unlike traditional real-valued approaches that often discard semantic phase information, the proposed supervised complex-valued contrastive learning (SCVCL) mechanism effectively synchronizes amplitude distributions with phase correlations, thereby enforcing rigorous intra-class compactness within the complex domain. Furthermore, by formally grounding the OOD state as the quantum empty set $\emptyset$ within a Hilbert space and leveraging the generalized quantum evidential combination rule (GQECR), the framework establishes a theoretically sound mechanism for quantifying

uncertainty and resolving conflicting evidence.

Despite these advancements, the framework currently faces limitations regarding the computational complexity inherent to ensemble-based complex arithmetic for feature reliability. Consequently, future research will prioritize the development of lightweight complex-valued fusion algorithms to optimize inference efficiency and the exploration of self-adaptive thresholding mechanisms to eliminate hyperparameter reliance. Additionally, we aim to extend this quantum evidential paradigm to heterogeneous multi-modal sensor fusion, thereby further enhancing the reliability and safety of autonomous recognition systems in unconstrained, OOD environments.

## Acknowledgments

This work was supported in part by the National Natural Science Foundation of China under Grant 62473067, in part by Xiaomi Young Talents Program, Chongqing Talents: Exceptional Young Talents Project under Grant cstc2022ycjh-bgzxm0070, in part by Chongqing Overseas Scholars Innovation Program under Grant cx2022024, and in part by Australian Research Council (ARC) Projects under Grant DE220100265, Grant DP250103612, and Grant LP240200636.The experimental and computational work in this research run on the Huawei Cloud AI Compute Service. We appreciate the stable compute supply from this platform. We are grateful to the Area Chair and the anonymous reviewers for their insightful comments and suggestions. Their feedback has helped us refine the presentation and strengthen the technical contributions of this work.

## Impact Statement

This paper presents work whose goal is to advance the field of Machine Learning. There are many potential societal consequences of our work, none which we feel must be specifically highlighted here.

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

# A. Preliminaries

## A.1. Complex-Valued Neural Network

Complex-Valued Neural Networks (CVNNs) (Trabelsi et al., 2017; Nguyen et al., 2022) generalize real-valued neural networks (RVNNs) to the complex field $\mathbb{C}$. Unlike RVNNs, which process amplitude and phase as independent channels, CVNNs treat a complex-valued data point $z = x + iy$ as a holistic entity. This algebraic structure allows the model to inherently preserve the phase-amplitude coupling, which is essential for signal modalities such as SAR.

The fundamental operation in a CVNN is the Complex Convolution. Let $\mathbf{h} = \mathbf{a} + i\mathbf{b}$ denote the complex input feature map and $\mathbf{W} = \mathbf{A} + i\mathbf{B}$ represent the complex convolution kernel, where $\mathbf{a}, \mathbf{b}$ are the real and imaginary parts of the input, and $\mathbf{A}, \mathbf{B}$ are the real and imaginary parts of the weights, respectively. The complex convolution output $\mathbf{z}$ is computed via the complex distributive law (Trabelsi et al., 2017; Nguyen et al., 2022):

$$\mathbf{z} = \mathbf{W} * \mathbf{h} = (\mathbf{A} * \mathbf{a} - \mathbf{B} * \mathbf{b}) + i(\mathbf{B} * \mathbf{a} + \mathbf{A} * \mathbf{b}). \tag{A.1}$$

This operation is implemented using a real-valued matrix form to enforce the weight-sharing mechanism characteristic of complex arithmetic (Trabelsi et al., 2017; Nguyen et al., 2022):

$$\begin{bmatrix} \Re(\mathbf{z}) \\ \Im(\mathbf{z}) \end{bmatrix} = \begin{bmatrix} \mathbf{A} & -\mathbf{B} \\ \mathbf{B} & \mathbf{A} \end{bmatrix} * \begin{bmatrix} \mathbf{a} \\ \mathbf{b} \end{bmatrix}. \tag{A.2}$$

To effectively transform real-valued raw data into the complex domain while capturing directional and frequency-selective features, we employ a Gabor Convolutional Layer as the initial encoding stage. This layer is formulated as a specialized instantiation of the complex convolution defined in Eq. (A.1), where the learnable weight matrices $\mathbf{A}$ and $\mathbf{B}$ are parameterized by the complex Gabor function.

A 2D complex Gabor kernel $G(x, y)$ is defined as (Trabelsi et al., 2017; Nguyen et al., 2022):

$$G(x, y) = \underbrace{\exp\left(-\frac{x'^2 + \gamma^2 y'^2}{2\sigma^2}\right)}_{\text{Envelope}} \cdot \exp\left(i\left(2\pi\frac{x'}{\lambda} + \psi\right)\right), \tag{A.3}$$

where $x' = x\cos\theta + y\sin\theta$ and $y' = -x\sin\theta + y\cos\theta$. By applying Euler's formula $e^{i\phi} = \cos\phi + i\sin\phi$, the Gabor kernel naturally decomposes into real and imaginary components. These components correspond directly to the weight matrices $\mathbf{A}$ and $\mathbf{B}$ in the complex convolution formulation (Trabelsi et al., 2017; Nguyen et al., 2022):

$$\mathbf{A}(x, y) = \Re(G(x, y)) = \exp\left(-\frac{x'^2 + \gamma^2 y'^2}{2\sigma^2}\right)\cos\left(2\pi\frac{x'}{\lambda} + \psi\right), \tag{A.4a}$$

$$\mathbf{B}(x, y) = \Im(G(x, y)) = \exp\left(-\frac{x'^2 + \gamma^2 y'^2}{2\sigma^2}\right)\sin\left(2\pi\frac{x'}{\lambda} + \psi\right). \tag{A.4b}$$

By constraining the convolutional weights $\mathbf{A}$ and $\mathbf{B}$ to these forms, the network acts as a learnable filter bank that explicitly extracts structural phase information from the outset, providing a robust physical foundation for subsequent processing.

Standard pooling operations are adapted via *Magnitude-based Max Pooling* (Trabelsi et al., 2017; Nguyen et al., 2022):

$$z_{out} = z_k, \quad \text{where } k = \arg\max_{j\in\mathcal{P}} |z_j|, \tag{A.5}$$

where $\mathcal{P}$ denotes the pooling region. Non-linearity is introduced via the split-activation function $\mathbb{C}\text{ReLU}(z) = \text{ReLU}(\Re(z)) + i\text{ReLU}(\Im(z))$ to avoid the constraints of Liouville's theorem. Finally, optimization employs Wirtinger calculus to compute gradients of the loss $\mathcal{L}$ with respect to complex parameters $\mathbf{W}$ (Trabelsi et al., 2017; Nguyen et al., 2022):

$$\nabla_{\mathbf{W}}\mathcal{L} = 2\frac{\partial\mathcal{L}}{\partial\mathbf{W}^*} = \frac{\partial\mathcal{L}}{\partial\mathbf{A}} + i\frac{\partial\mathcal{L}}{\partial\mathbf{B}}, \tag{A.6}$$

enabling standard backpropagation in the complex domain.

## A.2. Generalized Quantum Evidence Theory

Generalized quantum evidence theory (GQET) (Xiao, 2023; Aug. 2025; Yu et al., 2026; Xiao et al., 2025b) handles uncertainty in the open world by utilizing the generalized quantum mass function (GQMF) and addresses reasoning uncertainty through the QECR. In GQET, the quantum frame of discernment (QFOD) represents the likelihood of associated targets. The GQMF, which maps the QFOD to the quantum field, serves as a foundation for uncertainty modeling. Developing an appropriate form of the mass function is crucial to accurately modeling uncertainty.

**Definition A.1** (Quantum frame of discernment). A quantum frame of discernment (QFOD), denoted as $\Phi$, is defined as a collection of mutually exclusive and collectively exhaustive events. Each event is represented as an orthonormal basis $\phi_g$ within a Hilbert space, defined as follows (Xiao, 2023; Aug. 2025):

$$\Phi = \{\phi_1, \phi_2, \ldots, \phi_n\}. \tag{A.7}$$

**Definition A.2** (Quantum proposition). The power set of $\Phi$ is defined as (Xiao, 2023; Aug. 2025):

$$2^{\Phi} = \{\emptyset, \{\phi_1\}, \ldots, \{\phi_n\}, \{\phi_1\phi_2\}, \ldots, \{\phi_1\phi_2\ldots\phi_g\}, \ldots, \Phi\}, \tag{A.8}$$

where $\emptyset$ refers to an indeterminate event, simplified as:

$$2^{\Phi} = \{\emptyset, \phi_1, \ldots, \phi_n, \phi_{12}, \ldots, \phi_{12\ldots g}, \ldots, \phi_{12\ldots n}\}. \tag{A.9}$$

A quantum proposition $\psi_j$ is defined as any element $\psi_j \in 2^{\Phi}$.

**Definition A.3** (Generalized quantum mass function). A generalized quantum mass function (GQMF) $\mathbb{Q}_{\mathbb{M}}$ in QFOD $\Phi$, also called a GQBPA, is a mapping from $2^{\Phi}$ to $\mathbb{C}$, defined as (Xiao, 2023; Aug. 2025):

$$\mathbb{Q}_{\mathbb{M}} : 2^{\Phi} \to \mathbb{C}, \tag{A.10}$$

and satisfies:

$$\mathbb{Q}_{\mathbb{M}}(\psi_j) = \varphi(\psi_j)e^{i\theta(\psi_j)}, \quad \psi_j \in 2^{\Phi},$$
$$\sum_{\psi_j \in 2^{\Phi}} |\mathbb{Q}_{\mathbb{M}}(\psi_j)|^2 = 1, \tag{A.11}$$

where $i = \sqrt{-1}$, $\varphi(\psi_j) \in [0, 1]$ represents the magnitude of $\mathbb{Q}_{\mathbb{M}}(\psi_j)$, $\theta(\psi_j)$ is the phase of $\mathbb{Q}_{\mathbb{M}}(\psi_j)$, and $|\mathbb{Q}_{\mathbb{M}}(\psi_j)|^2$ is the squared amplitude of $\mathbb{Q}_{\mathbb{M}}(\psi_j)$, serving as the generalized quantum basic probability distribution (GQBPD).

**Definition A.4** (Generalized quantum basic probability distribution). The generalized quantum basic probability distribution (GQBPD) of $\mathbb{Q}_{\mathbb{M}}$ is defined as (Xiao, 2023; Aug. 2025):

$$\mathrm{M} : 2^{\Phi} \to [0, 1], \tag{A.12}$$

and satisfies:

$$\mathrm{M}(\psi_j) = |\mathbb{Q}_{\mathbb{M}}(\psi_j)|^2, \quad \psi_j \in 2^{\Phi},$$
$$\sum_{\psi_j \in 2^{\Phi}} \mathrm{M}(\psi_j) = 1. \tag{A.13}$$

**Definition A.5** (Quantum focal element). For $\psi_j \in 2^{\Phi}$, if $|\mathbb{Q}_{\mathbb{M}}(\psi_j)|$ or $\varphi(\psi_j) > 0$, $\psi_j$ is called a focal element of GQMF. $|\mathbb{Q}_{\mathbb{M}}(\psi_j)|^2$ represents the observed degree of belief or support to $\psi_j$ (Xiao, 2023; Aug. 2025).

**Definition A.6** (Generalized quantum belief function in GQET). Let $\mathbb{Q}_{\mathbb{M}}$ be a GQBPA with proposition $\psi_j \in 2^{\Phi}$. A generalized quantum belief function GQBel for $\psi_j$ in GQET, mapping from $2^{\Phi}$ to $[0, 1]$, is defined by (Xiao, 2023; Aug. 2025):

$$\mathrm{GQBel}(\psi_j) = \begin{cases} \sum_{\psi_p \subseteq \psi_j} |\mathbb{Q}_{\mathbb{M}}(\psi_p)|^2, & \psi_j \neq \emptyset, \\ |\mathbb{Q}_{\mathbb{M}}(\psi_j)|^2, & \psi_j = \emptyset. \end{cases} \tag{A.14}$$

According to Eq. (A.13), Eq. (A.14) can also be represented as:

$$\mathrm{GQBel}(\psi_j) = \begin{cases} \sum_{\psi_p \subseteq \psi_j} \mathrm{M}(\psi_p), & \psi_j \neq \emptyset, \\ \mathrm{M}(\psi_j), & \psi_j = \emptyset. \end{cases} \tag{A.15}$$

Therefore, when $\mathrm{M} = m_G$, the equation becomes:

$$\mathrm{GQBel}(\psi_j) = \begin{cases} \sum\limits_{\psi_p \subseteq \psi_j} m_G(\psi_p), & \psi_j \neq \emptyset, \\ m_G(\psi_j), & \psi_j = \emptyset, \end{cases} \tag{A.16}$$

which is consistent with the classical GBel in GET (Xiao, Aug. 2025).

**Definition A.7** (Generalized quantum plausibility function in GQET). Let $\mathbb{Q}_{\mathbb{M}}$ be a GQBPA with proposition $\psi_j \in 2^\Phi$. A generalized quantum plausibility function GQPl for $\psi_j$ in GQET, mapping from $2^\Phi$ to $[0, 1]$, is defined by (Xiao, 2023; Aug. 2025):

$$\mathrm{GQPl}(\psi_j) = \begin{cases} \sum\limits_{\psi_p \cap \psi_j \neq \emptyset} |\mathbb{Q}_{\mathbb{M}}(\psi_p)|^2, & \psi_j \neq \emptyset, \\ |\mathbb{Q}_{\mathbb{M}}(\psi_j)|^2, & \psi_j = \emptyset. \end{cases} \tag{A.17}$$

According to Eq. (A.13), Eq. (A.17) can also be represented as:

$$\mathrm{GQPl}(\psi_j) = \begin{cases} \sum\limits_{\psi_p \cap \psi_j \neq \emptyset} \mathrm{M}(\psi_p), & \psi_j \neq \emptyset, \\ \mathrm{M}(\psi_j), & \psi_j = \emptyset. \end{cases} \tag{A.18}$$

Therefore, when $\mathrm{M} = m_G$, the equation becomes:

$$\mathrm{GQPl}(\psi_j) = \begin{cases} \sum\limits_{\psi_p \cap \psi_j \neq \emptyset} m_G(\psi_p), & \psi_j \neq \emptyset, \\ m_G(\psi_j), & \psi_j = \emptyset, \end{cases} \tag{A.19}$$

which is consistent with the classical quantum plausibility function in quantum evidence theory (Xiao, Aug. 2025).

**Definition A.8** (Generalized quantum evidential combination rule (GQECR)). Consider a collection of independent GQBPAs, which is represented as $\{\mathbb{Q}_{\mathrm{M}1}, \cdots, \mathbb{Q}_{\mathrm{M}h}, \cdots, \mathbb{Q}_{\mathrm{M}k}\}$. Here, these GQBPAs are associated with the proposition $\psi_j$ within QFOD denoted as $\Phi$. The quantum evidential combination rule, symbolically expressed as $\mathbb{Q}_{\mathrm{M}1} \oplus \cdots \oplus \mathbb{Q}_{\mathrm{M}h} \oplus \cdots \oplus \mathbb{Q}_{\mathrm{M}k}$, is characterized and defined in the following manner (Xiao, 2023; Aug. 2025):

$$\mathbb{Q}_{\mathrm{M}1} \oplus \cdots \oplus \mathbb{Q}_{\mathrm{M}h} \oplus \cdots \oplus \mathbb{Q}_{\mathrm{M}k}(\psi_j) = \frac{\left| \sum\limits_{\cap \psi_p = \psi_j} \prod\limits_{1 \leq h \leq k} \mathbb{Q}_{\mathrm{M}h}(\psi_p) \right|^2}{\sum\limits_{\psi_p \subseteq \Phi} \left| \sum\limits_{\cap \psi_p = \psi_j} \prod\limits_{1 \leq h \leq k} \mathbb{Q}_{\mathrm{M}h}(\psi_p) \right|^2 + \left| \prod\limits_{1 \leq h \leq k} \mathbb{Q}_{\mathrm{M}h}(\emptyset) \right|^2}, \tag{A.20}$$

$$\mathbb{Q}_{\mathrm{M}1} \oplus \cdots \oplus \mathbb{Q}_{\mathrm{M}h} \oplus \cdots \oplus \mathbb{Q}_{\mathrm{M}k}(\emptyset) = \frac{\left| \prod\limits_{1 \leq h \leq k} \mathbb{Q}_{\mathrm{M}h}(\emptyset) \right|^2}{\sum\limits_{\psi_p \subseteq \Phi} \left| \sum\limits_{\cap \psi_p = \psi_j} \prod\limits_{1 \leq h \leq k} \mathbb{Q}_{\mathrm{M}h}(\psi_p) \right|^2 + \left| \prod\limits_{1 \leq h \leq k} \mathbb{Q}_{\mathrm{M}h}(\emptyset) \right|^2}. \tag{A.21}$$

### A.3. Quantum Evidence Theory

In the case where $\mathbb{Q}_{\mathrm{M}}(\emptyset) = 0$, indicating a closed world, the GQET reduces to quantum evidence theory (QET).

**Definition A.9** (Quantum basic probability amplitude function). A quantum basic probability amplitude (QBPA) function $\mathbb{Q}_{\mathrm{M}}$ in QFOD $\emptyset$, also referred to as a quantum mass function, is defined as a mapping (Xiao, 2023; Aug. 2025):

$$\mathbb{Q}_{\mathbb{M}} : 2^\Phi \to \mathbb{C}, \tag{A.22}$$

and satisfies (Xiao, 2023; Aug. 2025):

$$\mathbb{Q}_{\mathrm{M}}(\emptyset) = 0 \quad \text{and} \quad \mathbb{Q}_{\mathbb{M}}(\psi_j) = \varphi(\psi_j) e^{i\theta(\psi_j)}, \quad \psi_p \subseteq \Phi,$$
$$\sum\limits_{\psi_p \subseteq \Phi} |\mathbb{Q}_{\mathbb{M}}(\psi_j)|^2 = 1. \tag{A.23}$$

**Definition A.10** (Quantum basic probability function). The quantum basic probability function of $\mathbb{Q}_M$, also referred as a quantum basic probability distribution (QBPD), is defined as (Xiao, 2023; Aug. 2025):

$$M : 2^\Phi \to [0, 1], \tag{A.24}$$

and satisfies (Xiao, 2023; Aug. 2025):

$$M(\emptyset) = 0 \quad \text{and} \quad M(\psi_j) = |\mathbb{Q}_\mathbb{M}(\psi_j)|^2, \quad \psi_p \subseteq \Phi,$$
$$\sum_{\psi_p \subseteq \Phi} M(\psi_j) = 1. \tag{A.25}$$

**Definition A.11** (Quantum evidential combination rule). The quantum evidential combination rule (QECR), denoted as $\mathbb{Q}_{M1} \oplus \cdots \oplus \mathbb{Q}_{Mh} \oplus \cdots \oplus \mathbb{Q}_{Mk}(\psi_j)$, is defined as (Xiao, 2023; Aug. 2025):

$$\mathbb{Q}_{M1} \oplus \cdots \oplus \mathbb{Q}_{Mh} \oplus \cdots \oplus \mathbb{Q}_{Mk}(\psi_j) = \frac{\left| \sum_{\cap \psi_p = \psi_j} \prod_{1 \le h \le k} \mathbb{Q}_{Mh}(\psi_p) \right|^2}{\sum_{\psi_p \subseteq \Phi} \left| \sum_{\cap \psi_p = \psi_j} \prod_{1 \le h \le k} \mathbb{Q}_{Mh}(\psi_p) \right|^2}, \tag{A.26}$$

$$\mathbb{Q}_{M1} \oplus \cdots \oplus \mathbb{Q}_{Mh} \oplus \cdots \oplus \mathbb{Q}_{Mk}(\emptyset) = 0. \tag{A.27}$$

# B. Theoretical Analysis of Phase Projection Vector

In the Sec. 2.3, the Phase Projection Vector $\mathbf{p}_t$ is introduced to handle complex-valued phase information within a contrastive learning framework. This section provides the mathematical justification for this transformation, specifically addressing the *Cyclic Ambiguity Problem* and proving the *Isometry Property* between the projected Euclidean space and the native circular topology.

## B.1. The Cyclic Ambiguity Problem

Let $\theta \in [-\pi, \pi]$ represent the phase of a complex number. Standard metric learning approaches typically employ the $L_p$ norm (e.g., Euclidean distance or L1 distance) on raw feature vectors. However, applying these metrics directly to raw phase values leads to a topological violation ID as cyclic ambiguity.

Consider two phase angles $\theta_1 = \pi - \epsilon$ and $\theta_2 = -\pi + \epsilon$, where $\epsilon \to 0^+$.

- **Geometrically on the unit circle:** These two points are adjacent, with a circular distance approaching 0.

- **Numerically in Euclidean space:** The distance is $|(\pi - \epsilon) - (-\pi + \epsilon)| = |2\pi - 2\epsilon| \approx 2\pi$.

This discrepancy creates a massive "loss cliff" during backpropagation. If the network pushes a prediction from $\pi$ slightly over the boundary to $-\pi$, a standard loss function operating on raw angles would penalize this as a maximum error rather than a minimal update.

## B.2. Derivation of the Isometry Property

To resolve the ambiguity described above, the phase space $\mathbb{T}^D$ (where $\mathbb{T}$ is the torus) is mapped to a Euclidean space $\mathbb{R}^{2D}$ via the mapping $\Psi : [-\pi, \pi]^D \to \mathbb{R}^{2D}$.

Let $\boldsymbol{\theta}_t = [\theta_t^{(1)}, \ldots, \theta_t^{(D)}]^\top$ be the raw phase vector. The projection $\mathbf{p}_t = \Psi(\boldsymbol{\theta}_t)$ is defined as:

$$\mathbf{p}_t = \left[ \cos \theta_t^{(1)}, \sin \theta_t^{(1)}, \ldots, \cos \theta_t^{(D)}, \sin \theta_t^{(D)} \right]^\top. \tag{A.28}$$

**Proposition B.1.** *The cosine similarity between two phase projection vectors $\mathbf{p}_t$ and $\mathbf{p}_j$ in $\mathbb{R}^{2D}$ is mathematically equivalent to the mean cosine of the angular differences in the original phase domain.*

The cosine similarity is defined as:

$$\text{sim}(\mathbf{p}_t, \mathbf{p}_j) = \frac{\mathbf{p}_t^\top \mathbf{p}_j}{\|\mathbf{p}_t\|_2 \|\mathbf{p}_j\|_2}. \tag{A.29}$$

**Step 1: Analyze the Norm.** First, the $L_2$ norm of any projection vector $\mathbf{p}_t$ is computed. Since $\mathbf{p}_t$ consists of $D$ pairs of sine and cosine components:

$$\|\mathbf{p}_t\|_2 = \sqrt{\sum_{k=1}^{D} \left( (\cos \theta_t^{(k)})^2 + (\sin \theta_t^{(k)})^2 \right)} = \sqrt{\sum_{k=1}^{D} 1} = \sqrt{D}. \tag{A.30}$$

Thus, the magnitude of the projection vector is constant regardless of the phase values.

**Step 2: Analyze the Dot Product.** Next, the dot product $\mathbf{p}_t^\top \mathbf{p}_j$ is computed:

$$\mathbf{p}_t^\top \mathbf{p}_j = \sum_{k=1}^{D} \left( \cos \theta_t^{(k)} \cos \theta_j^{(k)} + \sin \theta_t^{(k)} \sin \theta_j^{(k)} \right). \tag{A.31}$$

By applying the trigonometric difference identity $\cos(A - B) = \cos A \cos B + \sin A \sin B$, this simplifies to:

$$\mathbf{p}_t^\top \mathbf{p}_j = \sum_{k=1}^{D} \cos \left( \theta_t^{(k)} - \theta_j^{(k)} \right). \tag{A.32}$$

**Step 3: Combine Results.** Substituting the results from Step 1 and Step 2 into the similarity formula:

$$
\begin{aligned}
\text{sim}(\mathbf{p}_t, \mathbf{p}_j) &= \frac{\sum_{k=1}^{D} \cos(\theta_t^{(k)} - \theta_j^{(k)})}{\sqrt{D} \cdot \sqrt{D}} \\
&= \frac{1}{D} \sum_{k=1}^{D} \cos \left( \theta_t^{(k)} - \theta_j^{(k)} \right).
\end{aligned}
\tag{A.33}
$$

### B.3. Optimization Stability Analysis

The derived equivalence $\text{sim}(\mathbf{p}_t, \mathbf{p}_j) \propto \sum \cos(\Delta\theta)$ has profound implications for optimization stability:

1. **Periodicity:** The function $f(\Delta\theta) = \cos(\Delta\theta)$ is naturally $2\pi$-periodic. Hence, $\cos(\pi - (-\pi)) = \cos(2\pi) = 1$, correctly identifying the boundary condition as a perfect match.

2. **Gradient Smoothness:** The gradient of the loss with respect to the raw phase $\theta$ behaves as $\frac{\partial \mathcal{L}}{\partial \text{sim}} \cdot (-\sin(\Delta\theta))$. This gradient is bounded and continuous everywhere. In contrast, a loss based on raw Euclidean distance $L = \|\theta_t - \theta_j\|$ has a gradient discontinuity at the periodicity boundary, leading to unstable training dynamics for complex-valued neural networks.

Therefore, the construction of $\mathbf{p}_t$ is not merely a feature expansion but a necessary geometric transformation to ensure the mathematical validity of contrastive learning in the complex domain.

## C. Theoretical Justification of Generalized Quantum Basic Probability Amplitude Mapping Function

In Section 2, the Quantum Probability Normalization Mapping utilizes a specific amplification factor $\xi(\cdot)$ to transform the generalized complex-valued feature vector $\tilde{\mathbf{f}}_v$ into valid generalized quantum basic probability amplitude (GQBPA). This section provides the theoretical derivation and justification for this design, demonstrating how it bridges the discriminative power of the Softmax function with the phase-preserving requirements of generalized quantum evidence theory (GQET).

## C.1. The Dilemma of Complex-Valued Probability Assignment

Let $\mathbf{z} \in \mathbb{C}^K$ be a complex-valued feature vector. In standard real-valued classification, the probability distribution is typically obtained via the Softmax function on the logits' magnitudes:

$$P(y = k) = \text{Softmax}(|\mathbf{z}|)_k = \frac{e^{|z^{(k)}|}}{\sum_j e^{|z^{(j)}|}}. \tag{A.34}$$

The exponential mapping $x \mapsto e^x$ in Softmax is crucial because it sharpens the decision boundary and suppresses low-confidence classes, effectively pushing the mass toward the dominant class.

However, in the proposed TrustworthyQENN framework, the fusion of evidence (via GQECR) must occur *before* the final probability decision to utilize quantum interference.

- **Problem 1 (Phase Loss):** Direct application of Eq. (A.34) results in a real number, causing the permanent loss of critical phase information $\theta = \arg(z)$ and rendering quantum interference impossible.

- **Problem 2 (Lack of Discriminability):** If a linear normalization is employed (e.g., $z^{(k)}/\sum |z^{(j)}|$), the magnitude distribution remains linear. Without the exponential expansion, the margin between the true class and the OOD class (represented by $|\emptyset\rangle$) is insufficient, leading to high entropy in the evidence and reduced rejection performance.

Therefore, a mapping $\Gamma : \mathbb{C} \to \mathbb{C}$ is sought that satisfies two conditions simultaneously:

1. **Phase Consistency:** $\arg(\Gamma(z)) = \arg(z)$.

2. **Magnitude Sharpening:** $|\Gamma(z)| = e^{|z|}$.

## C.2. Derivation of the Amplification Factor

**Proposition C.1.** *The quantum probability amplification factor $\xi(z) = \frac{e^{|z|}}{|z|}$ is the unique scalar multiplier that transforms a complex number $z$ such that its magnitude scales exponentially while its phase remains invariant.*

*Proof.* Let the transformed complex value be denoted as $q = \Gamma(z)$. To preserve phase information (Condition 1), the transformation must be a scaling of the original vector $z$ by a real-valued scalar $\xi \in \mathbb{R}^+$:

$$q = z \cdot \xi. \tag{A.35}$$

Taking the modulus of both sides:

$$|q| = |z| \cdot |\xi|. \tag{A.36}$$

Since $\xi$ is a positive scaling factor, $|\xi| = \xi$. To satisfy the magnitude sharpening requirement (Condition 2), it is imposed that the magnitude of the transformed evidence must follow the exponential behavior of logits in a Softmax function:

$$|q| \equiv e^{|z|}. \tag{A.37}$$

Substituting this into the modulus equation:

$$e^{|z|} = |z| \cdot \xi. \tag{A.38}$$

Solving for $\xi$ yields:

$$\xi = \frac{e^{|z|}}{|z|}. \tag{A.39}$$

Thus, the operation $\tilde{\mathbf{f}}_v^{(j)} \cdot \xi(\tilde{\mathbf{f}}_v^{(j)})$ produces a complex evidence mass with magnitude $e^{|\tilde{\mathbf{f}}_v^{(j)}|}$ and phase $\arg(\tilde{\mathbf{f}}_v^{(j)})$. $\qquad\square$

# D. datasets and Experimental Setup Details

## D.1. Dataset Details

To ensure a unified standard, for each dataset, the classes are partitioned into a set of *ID classes* ($\mathcal{Y}_{in}$) and *OOD classes* ($\mathcal{Y}_{out}$). The model is trained exclusively on samples from $\mathcal{Y}_{in}$. During testing, the model is evaluated on samples from both $\mathcal{Y}_{in}$ and $\mathcal{Y}_{out}$. The openness of the task is defined as:

$$\mathbb{O} = 1 - \sqrt{\frac{2 \times |\mathcal{Y}_{in}|}{|\mathcal{Y}_{in}| + |\mathcal{Y}_{out}|}} \tag{A.40}$$

Detailed descriptions of the three datasets are as follows:

- **MSTAR** (Xiao et al., 2025c): A benchmark SAR dataset for ground target recognition consisting of X-band SAR imagery covering varying depression angles. 3 classes are randomly selected as $\mathcal{Y}_{in}$, while the remaining 7 classes are designated as $\mathcal{Y}_{out}$, resulting in $\mathbb{O} = 32.06\%$.

- **EuroSAT** (Helber et al., 2019): A dataset comprising multi-spectral satellite images. While primarily optical, the multi-channel input is treated to simulate complex interactions or transformed into the complex domain to evaluate generalizability. 3 classes are randomly selected as $\mathcal{Y}_{in}$, while the remaining 7 classes are designated as $\mathcal{Y}_{out}$, resulting in $\mathbb{O} = 32.06\%$.

- **FUSAR-Ship** (Wang et al., 2022a): A high-resolution SAR ship detection dataset. The complex backscattering characteristics of ships against the sea clutter background present significant challenges for OOD. 3 classes are randomly selected as $\mathcal{Y}_{in}$, while the remaining 7 classes are designated as $\mathcal{Y}_{out}$, resulting in $\mathbb{O} = 32.06\%$.

## D.2. Implementation and Hyperparameters

**Network Architecture:** Based on the structure described in (Xu et al., 2023), we designed TrustworthyQENN, a modular complex-valued neural network comprising a shared encoder, a projection network, and a classifier head. All complex-valued operations – including convolution, batch normalization, and activation functions – are implemented using a PyTorch-compatible complex-valued library. The encoder initiates with a learnable Gabor layer sourced from (Noé et al., 2020), which transforms raw input data into the complex domain. This is followed by eight convolutional blocks utilizing complex-valued Batch Normalization ($\mathbb{C}$BN) and split-$\mathbb{C}$ReLU activation functions to ensure optimization stability. Following the encoder, the projection network maps latent features into a 128-dimensional space for contrastive learning, while the classifier head generates the final discriminative features required for quantum evidential fusion. The detailed configuration of the architecture is summarized in Table A.1.

**Training settings:** The input images are resized to $64 \times 64$. The network is trained using the Adam optimizer with an initial learning rate of $1 \times 10^{-3}$ and a weight decay of $1 \times 10^{-4}$. The learning rate is decayed by a factor of 0.1 every 30 epochs (Liu et al., 2022). The temperature parameter $\tau$ is empirically set at 0.1 (Xu et al., 2023). The threshold $\delta$ is adopted as the same setting specified in (Yoshihashi et al., 2019). Consistent with (Wang et al., 2025), the ensemble size $N$ is set to 5. All results for non-ensemble experiments are reported as the average of 10 independent trials, whereas the results for ensemble models are calculated as the mean of all possible $\binom{10}{5}$ combinations of 5-model ensembles derived from 10 independent trials.

**Evaluation metrics:** To comprehensively measure OSR performance, three standard metrics are employed from (Liu et al., 2022; Bendale & Boult, 2016): 1) AUROC (Area Under the Receiver Operating Characteristic curve), which measures the overall separation between ID and OOD distributions; 2) OSCR (Open Set Classification Rate), which considers both correct classification of ID classes and detection of OOD classes; 3) FPR95 (False Positive Rate at 95% True Positive Rate), where a lower value indicates fewer OOD samples are misclassified as ID; and 4) ACC (Closed Set Accuracy), which is used to evaluate the ability of OOD detection methods to maintain robust classification performance on ID classes.

## D.3. Comparative Methods Details

The details of the comparative methods used in the main experiments are presented as follows:

- Softmax: A baseline closed-set classifier using maximum probability thresholding for rejection.

*Table A.1.* Detailed architecture of the proposed TrustworthyQENN. The network consists of a shared complex-valued encoder (ComplexBaseCNN), a projection net for contrastive learning, and a classifier head for recognition. All operations are performed in the complex domain ($\mathbb{C}$). Abbreviations: $Ke$: Kernel size, $St$: Stride, $Pa$: Padding, $Ch$: Output Channels.

| Module | Layer | Configuration ($Ke \times Ke, Ch, St, Pa$) | Output Size |
|--------|-------|---------------------------------------------|-------------|
| **Encoder $E(\cdot)$** | Input | - | $3 \times 64 \times 64$ |
| | Gabor Block | Gabor$\mathbb{C}$Conv $(5 \times 5, 64, 1, 2) + \mathbb{C}$BN2d $+ \mathbb{C}$ReLU | $64 \times 64 \times 64$ |
| | $\mathbb{C}$Conv Block 1 | $\mathbb{C}$Conv $(3 \times 3, 64, 1, 1) + \mathbb{C}$BN2d $+ \mathbb{C}$ReLU | $64 \times 64 \times 64$ |
| | $\mathbb{C}$Conv Block 2 | $\mathbb{C}$Conv $(3 \times 3, 128, 2, 1) + \mathbb{C}$BN2d $+ \mathbb{C}$ReLU | $128 \times 32 \times 32$ |
| | $\mathbb{C}$Conv Block 3 | $\mathbb{C}$Conv $(3 \times 3, 128, 1, 1) + \mathbb{C}$BN2d $+ \mathbb{C}$ReLU | $128 \times 32 \times 32$ |
| | $\mathbb{C}$Conv Block 4 | $\mathbb{C}$Conv $(3 \times 3, 128, 1, 1) + \mathbb{C}$BN2d $+ \mathbb{C}$ReLU | $128 \times 32 \times 32$ |
| | $\mathbb{C}$Conv Block 5 | $\mathbb{C}$Conv $(3 \times 3, 128, 2, 1) + \mathbb{C}$BN2d $+ \mathbb{C}$ReLU | $128 \times 16 \times 16$ |
| | $\mathbb{C}$Conv Block 6 | $\mathbb{C}$Conv $(3 \times 3, 128, 1, 1) + \mathbb{C}$BN2d $+ \mathbb{C}$ReLU | $128 \times 16 \times 16$ |
| | $\mathbb{C}$Conv Block 7 | $\mathbb{C}$Conv $(3 \times 3, 128, 1, 1) + \mathbb{C}$BN2d $+ \mathbb{C}$ReLU | $128 \times 16 \times 16$ |
| | $\mathbb{C}$Conv Block 8 | $\mathbb{C}$Conv $(3 \times 3, 128, 2, 1) + \mathbb{C}$BN2d $+ \mathbb{C}$ReLU | $128 \times 8 \times 8$ |
| | Global Pooling | $\mathbb{C}$AvgPool $(8 \times 8, 1, 0) + \mathbb{C}$Flatten | 128 |
| **Projection $G(\cdot)$** | $\mathbb{C}$Linear 1 | $\mathbb{C}$Linear $(128 \rightarrow 128) + \mathbb{C}$BN1d $+ \mathbb{C}$ReLU | 128 |
| | $\mathbb{C}$Linear 2 | $\mathbb{C}$Linear $(128 \rightarrow 128) + \mathbb{C}$BN1d | 128 |
| **Classifier $C(\cdot)$** | $\mathbb{C}$Linear 1 | $\mathbb{C}$Linear $(128 \rightarrow 128) + \mathbb{C}$BN1d $+ \mathbb{C}$ReLU | 128 |
| | $\mathbb{C}$Linear 2 | $\mathbb{C}$Linear $(128 \rightarrow |L_{kn}|) + \mathbb{C}$BN1d | $|L_{kn}|$ |

- OpenMax(Bendale & Boult, 2016): A seminal method using Extreme Value Theory (EVT) to calibrate prediction scores for OOD probability estimation.

- ODL / ODL+(Liu et al., 2022): Methods optimizing orientational spatial feature distributions via hierarchical attention; ODL+ extends this to a composite multi-layer feature space.

- ARPL(Chen et al., 2021): An Adversarial Reciprocal Point Learning (ARPL) framework minimizing both empirical and open space risks via reciprocal points and adversarial margins.

- ConOSR(Xu et al., 2023): A contrastive framework utilizing mixup-generated virtual outliers to refine representations by contrasting knowns against synthesized unknowns.

- ASC-RPL(Xiao et al., 2025c): A method integrating Attribute Scattering Center (ASC) features with Reciprocal Point Learning (RPL) to bound open space.

- DEF(Li et al., 2025): A neural collapse inspired method using Dual Equiangular Tight Frame(DEF) loss to align features with a simplex equiangular tight frame for optimal separation.

- CW(Wang et al., 2025): An uncertainty estimation strategy constructing credal sets from model averaging to better capture uncertainty.

# E. Additional Experiments

### E.1. Additional Cross-Dataset Analysis

To provide a holistic view of the model's robustness, we supplement the AUROC analysis in the main text with detailed comparisons of OSCR and FPR95 metrics under reciprocal cross-dataset settings. As illustrated in Fig. A.1, TrustworthyQENN consistently demonstrates superior performance regardless of which dataset serves as the ID source. In terms of OSCR, which measures the trade-off between closed-set accuracy and open-set rejection, TrustworthyQENN outperforms the runner-up method CW in both scenarios (Fig. A.1(a) and (b)). This indicates that our method effectively maintains high classification precision for known targets while suppressing the confidence of OOD samples. More notably, regarding the FPR95 metric (Fig. A.1(c) and (d)), the performance margin of TrustworthyQENN becomes even more pronounced.

Specifically, in the challenging scenario where FUSAR-Ship is the ID dataset (Fig. A.1(d)), TrustworthyQENN significantly lowers the false positive rate compared to real-valued baselines like ConOSR. This validates that the proposed SCVCL mechanism successfully learns invariant geometric semantics from complex-valued data, enabling the model to distinguish between different sensor modalities (MSTAR vs. FUSAR-Ship) with high reliability.

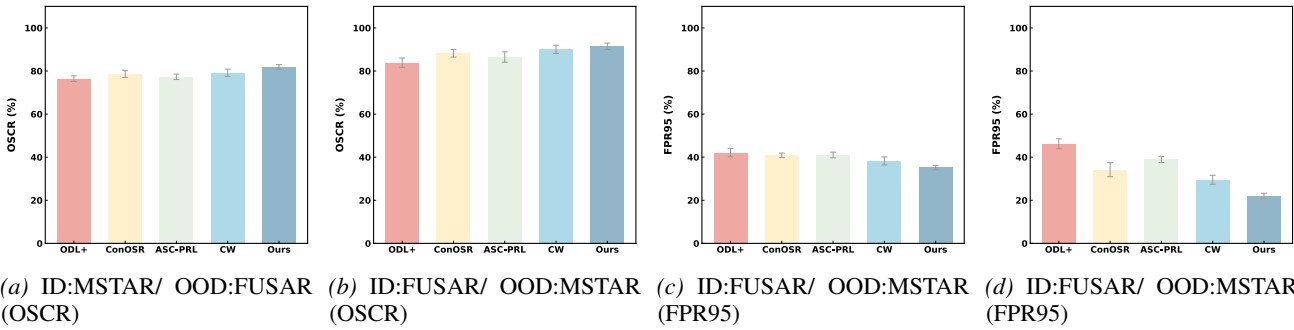

*(a)* ID:MSTAR/ OOD:FUSAR (OSCR)   *(b)* ID:FUSAR/ OOD:MSTAR (OSCR)   *(c)* ID:FUSAR/ OOD:MSTAR (FPR95)   *(d)* ID:FUSAR/ OOD:MSTAR (FPR95)

*Figure A.1.* Performance comparison of OOD metrics under reciprocal ID and OOD configurations between MSTAR and FUSAR-Ship datasets.

## E.2. Additional Sensitivity Experiment

To supplement the sensitivity analysis presented in Section 3.5, which primarily focused on the MSTAR dataset, this subsection provides the corresponding results for the EuroSAT and FUSAR-ship datasets (see Fig. A.2).

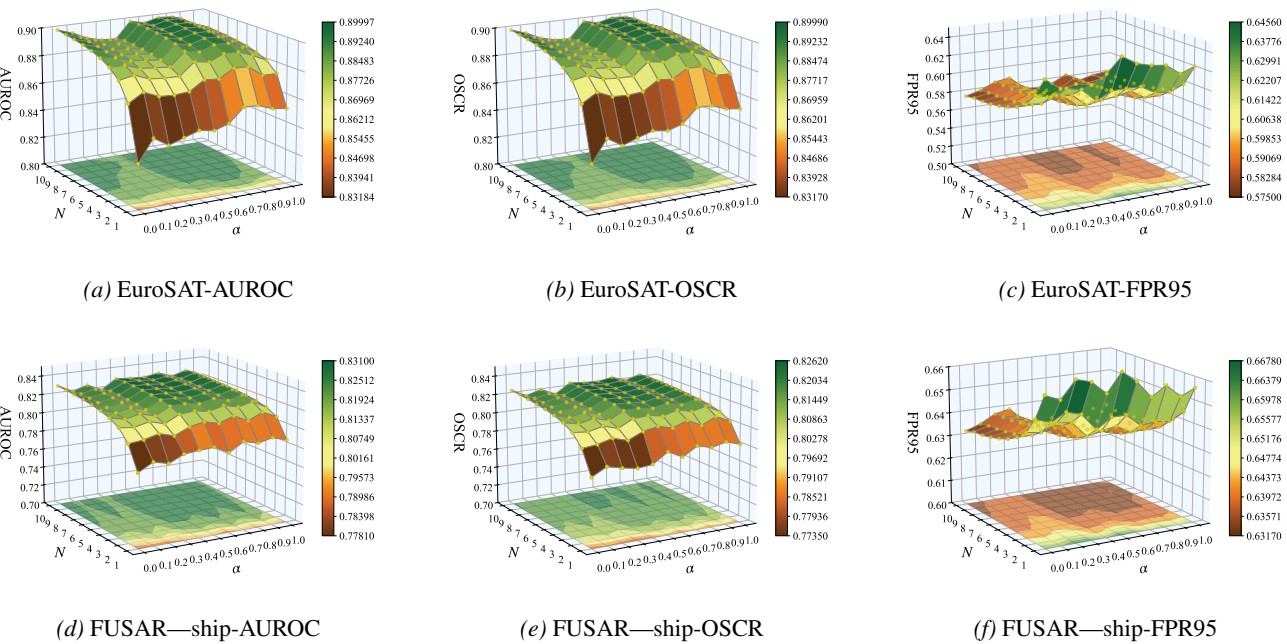

*(a)* EuroSAT-AUROC   *(b)* EuroSAT-OSCR   *(c)* EuroSAT-FPR95

*(d)* FUSAR—ship-AUROC   *(e)* FUSAR—ship-OSCR   *(f)* FUSAR—ship-FPR95

*Figure A.2.* Sensitivity analysis of the TrustworthyQENN on the EuroSAT and FUSAR—ship dataset with respect to the balance parameter $\alpha$ and ensemble size $N$.

## E.3. Performance under Varying Openness

To evaluate the robustness of TrustworthyQENN against varying degrees of "OOD-ness," experiments are conducted across six distinct scenarios (A through F) using the MSTAR dataset. As detailed in Table A.2, the openness $\mathbb{O}$ is modulated by progressively increasing the number of ID classes $|\mathcal{Y}_{in}|$ while decreasing the number of OOD classes $|\mathcal{Y}_{out}|$. The comparative results across these scenarios for TrustworthyQENN, CW, ConOSR, and OpenMax are illustrated in Fig A.3. It is observed that TrustworthyQENN maintains superior performance stability as openness increases. In high-openness

scenarios (Scenario A and B), TrustworthyQENN exhibits a significant margin over baseline methods in AUROC and OSCR. While all methods converge slightly as the task approaches a near-closed-set problem (Scenario F), TrustworthyQENN consistently records the lowest FPR95, demonstrating its precision in bounding the open space.

*Table A.2.* Composition of ID and OOD classes across varying openness scenarios in MSTAR.

| Scenario | ID Class IDs | OOD Class IDs | Openness ($\mathbb{O}$) |
|---|---|---|---|
| A | 0, 1, 2 | 3, 4, 5, 6, 7, 8, 9 | 32.06% |
| B | 0, 1, 2, 3 | 4, 5, 6, 7, 8, 9 | 24.41% |
| C | 0, 1, 2, 3, 4 | 5, 6, 7, 8, 9 | 18.35% |
| D | 0, 1, 2, 3, 4, 5 | 6, 7, 8, 9 | 13.40% |
| E | 0, 1, 2, 3, 4, 5, 6 | 7, 8, 9 | 9.25% |
| F | 0, 1, 2, 3, 4, 5, 6, 7 | 8, 9 | 5.72% |

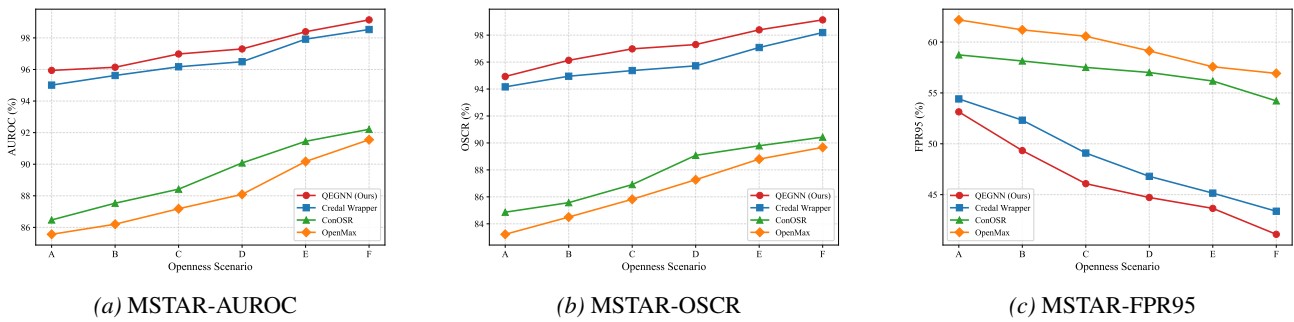

| (a) MSTAR-AUROC | (b) MSTAR-OSCR | (c) MSTAR-FPR95 |
|---|---|---|

*Figure A.3.* Performance comparison across different openness scenarios (A–F) on the MSTAR dataset. TrustworthyQENN demonstrates superior robustness as the number of OOD classes increases.

### E.4. Confusion Matrix Analysis

To evaluate the fine-grained performance across ID and OOD categories, we analyze the confusion matrices on the MSTAR dataset (see Fig. A.4), where 'U' denotes the OOD class. Compared to the baselines, TrustworthyQENN demonstrates a significantly stronger capability in identifying OOD samples, correctly rejecting 1770 samples, which is markedly higher than OpenMax (1459), ConOSR (1693), and CW (1749). Furthermore, TrustworthyQENN exhibits the lowest leakage of OOD samples into ID classes (0, 1, and 2) and maintains the highest diagonal accuracy for ID classes.

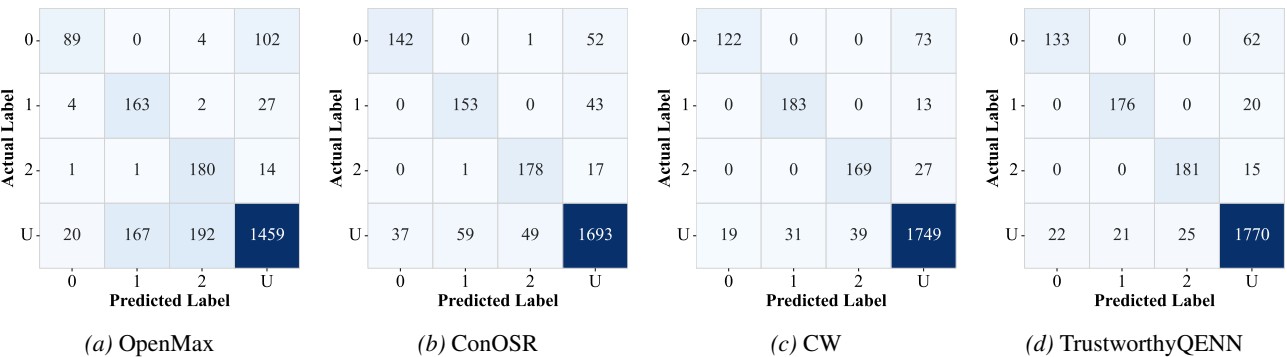

| (a) OpenMax | (b) ConOSR | (c) CW | (d) TrustworthyQENN |
|---|---|---|---|

*Figure A.4.* Confusion matrices of (a) OpenMax, (b) ConOSR, (c) CW, and (d) TrustworthyQENN.

### E.5. Parameter Efficiency Analysis

A common critique regarding Complex-Valued Neural Networks (CVNNs) is that their superior performance might stem solely from the doubled parameter count (real and imaginary parts) rather than the algebraic structure itself. To decouple

the impact of model capacity from the proposed SCVCL mechanism, a rigorous comparative experiment was conducted against the real-valued baseline, ConOSR. Specifically, the standard ConOSR was evaluated against two capacity-augmented variants—one with doubled channel widths and another with doubled depth—and compared to an SCVCL model with halved channel dimensions, which maintains a parameter budget roughly equivalent to the standard real-valued ConOSR. As summarized in Table A.3, simply increasing the capacity of the real-valued model (ConOSR $2\times$ Channels/Layers) yields only marginal improvements (e.g., MSTAR AUROC increases from 86.47% to 87.52%). In contrast, the SCVCL ($0.5\times$ Channels) significantly outperforms these baselines with an AUROC of 89.15%, despite having a comparable parameter count. This empirical evidence strongly suggests that the performance gain is not a trivial product of increased parameters, but is attributed to the mechanism's ability to capture the intrinsic phase-amplitude coupling and the physical geometry of the SAR data, which real-valued networks fail to model regardless of their size.

*Table A.3.* Parameter efficiency analysis comparing SCVCL with capacity-augmented real-valued baselines. The best result is **bolded**, and the second best is underlined.

| Method | MSTAR | | | EuroSAT | | | FUSAR-Ship | | |
|---|---|---|---|---|---|---|---|---|---|
| | AUROC↑ | OSCR↑ | FPR95↓ | AUROC↑ | OSCR↑ | FPR95↓ | AUROC↑ | OSCR↑ | FPR95↓ |
| ConOSR (Baseline) | 86.47 | 84.86 | 58.74 | 83.00 | 81.20 | 61.39 | 75.50 | 71.17 | 68.97 |
| ConOSR ($2\times$ Channels) | 87.52 | 85.91 | 57.30 | 84.15 | 82.05 | 60.12 | 76.24 | 72.33 | 68.10 |
| ConOSR ($2\times$ Layers) | 87.10 | 85.44 | 57.85 | 83.89 | 81.88 | 60.85 | 75.95 | 71.90 | 68.45 |
| SCVCL ($0.5\times$ Channels) | 89.15 | 89.02 | 57.10 | 83.95 | 83.85 | 59.95 | 77.45 | 76.80 | 70.15 |
| SCVCL | **91.24** | **91.19** | **56.60** | **84.16** | **84.13** | **59.58** | **79.36** | **78.85** | **69.86** |

*Table A.4.* Ablation study of different fusion strategies. GQECR outperforms both naive complex addition and classical evidential fusion (CW) applied to magnitudes. The best result is **bolded**, and the second best is underlined.

| Fusion Strategy | MSTAR | | | EuroSAT | | | FUSAR-Ship | | |
|---|---|---|---|---|---|---|---|---|---|
| | AUROC↑ | OSCR↑ | FPR95↓ | AUROC↑ | OSCR↑ | FPR95↓ | AUROC↑ | OSCR↑ | FPR95↓ |
| Complex Feature Addition | 92.85 | 92.50 | 56.40 | 86.40 | 86.10 | 58.75 | 80.12 | 79.50 | 67.71 |
| CW (Amplitude) | 95.01 | 94.16 | 54.41 | 88.10 | 87.95 | 58.20 | 81.05 | 80.44 | 67.15 |
| **TrustworthyQENN (GQECR)** | **95.94** | **95.93** | **53.14** | **89.53** | **89.50** | **57.14** | **82.67** | **82.19** | **66.23** |

*Table A.5.* OOD detection performance on standard visual benchmarks (ID: CIFAR-10). The best method is emphasized in **bold**.

| Method | OOD: SVHN | | | OOD: TinyImageNet | | |
|---|---|---|---|---|---|---|
| | AUROC↑ | OSCR↑ | FPR95↓ | AUROC↑ | OSCR↑ | FPR95↓ |
| ConOSR | 96.81 | 91.82 | 19.70 | 91.29 | 87.38 | 43.30 |
| CW | 97.46 | 93.71 | 14.33 | 93.10 | 90.32 | 41.02 |
| **Ours** | **98.94** | **96.09** | **10.86** | **94.92** | **92.83** | **37.29** |

### E.6. Fusion Strategies Analysis

To validate the necessity and effectiveness of the proposed GQECR, the method was benchmarked against two alternative fusion strategies within the TrustworthyQENN framework: 1) Complex Feature Addition, a naive approach where the complex-valued feature vectors from the ensemble are element-wise averaged before passing through the final classifier; and 2) CW, a hybrid approach where complex features are first converted to real-valued magnitudes (discarding phase) and then fused using the CW. The results in Table A.4 demonstrate the superiority of GQECR. While Complex Feature Addition retains phase information, its linear nature fails to effectively manage conflicting evidence from diverse views, resulting in sub-optimal rejection performance (FPR95: 56.40% on MSTAR). The CW improves upon naive addition by managing uncertainty, yet it falls short of GQECR (AUROC gap of 0.93% on MSTAR) due to the loss of semantic phase information during the magnitude conversion. GQECR effectively leverages both the uncertainty management of evidence theory and the semantic richness of complex phase, providing the most robust OOD detection.

### E.7. Evaluation on Standard Vision Benchmarks

To demonstrate that the proposed TrustworthyQENN framework is not exclusively constrained to remote sensing or SAR imagery, we further evaluate its generalization capability on standard visual benchmarks. Specifically, we utilize CIFAR-10 as the ID dataset, and select SVHN and TinyImageNet as the OOD datasets. This configuration rigorously tests the model's robustness against significant semantic shifts within standard optical image domains.

As summarized in Table A.5, TrustworthyQENN consistently outperforms the baseline contrastive method (ConOSR) and the state-of-the-art uncertainty-centric approach (CW). Notably, when evaluated against the SVHN dataset, our method achieves an AUROC of 98.94% (an improvement of 1.48% over CW) and reduces the FPR95 to 10.86%. Similarly, on the more challenging TinyImageNet OOD dataset, TrustworthyQENN achieves an AUROC of 94.92% (+1.82% over CW) while lowering the FPR95 to 37.29%. These results substantiate that the synergistic integration of complex-valued representation learning and generalized quantum evidential theory establishes a highly generalizable and robust framework. By mapping real-valued inputs into a complex-valued domain, the model significantly expands its representational capacity, generating highly discriminative features that effectively capture intricate spatial patterns.

