# OpenReview forum: "TrustworthyQENN: A Quantum Evidential Neural Network Based on Complex-Valued Contrastive Learning for Uncertainty Pattern Classification"
_ICML.cc/2026/Conference — ICML 2026 regular_

### Official Review · Reviewer_D78M · 2026-03-10

**Soundness:** 2
**Presentation:** 2
**Significance:** 2
**Originality:** 2
**Overall Recommendation:** 2
**Confidence:** 2

**Summary:**

This paper proposes a framework called TrustworthyQENN for Out-of-Distribution (OOD) detection, specifically for complex-valued data like SAR images. The method introduces Supervised Complex-Valued Contrastive Learning (SCVCL) to extract features from both the amplitude and phase of the data. Then, it maps these features into a Generalized Quantum Evidence Theory (GQET) framework, where OOD samples are treated as a "quantum empty set" ($\varnothing$). Finally, it fuses the outputs from multiple models to make the final prediction.

**Compliance With Llm Reviewing Policy:**

Affirmed.

**Key Questions For Authors:**

1. In Section 2.4.2, why is an OOD sample defined as the "quantum empty set ($\varnothing$)"? An OOD sample is a real image with features. How does an unknown but real object logically equal an empty or impossible state?

2. In Definition 2.8, the mapping function uses standard math like exponential scaling. What specific quantum properties (like superposition or entanglement) are actually used here? Why isn't a normal Softmax or probability scaling enough?

3. In Definition 2.2, mapping angles to sine and cosine is a standard way to handle cyclic data. What is the core new contribution of this specific step that makes it central to the proposed SCVCL?

**Limitations:**

The authors mentioned the high computing cost of their method in the conclusion. However, they did not discuss the limits of their theory. They should discuss the potential problems of forcing normal, everyday image data (like EuroSAT) into a strict quantum physics framework.

**Strengths And Weaknesses:**

**Strengths**:

1. Good experimental results: The paper tests the method on several datasets (MSTAR, EuroSAT, FUSAR-Ship). The numerical results are solid and show clear improvements over the baseline methods in different testing setups.

2. Practical motivation: Real-valued networks often ignore phase information. Trying to use both phase and amplitude to find OOD samples is a practical and useful direction, especially for radar and signal data.

---

**Weaknesses**:

1. In Section 2.4.2, the paper models the OOD state as the "quantum empty set ($\varnothing$)". In math and logic, an empty set means nothingness or an impossible event. However, an OOD sample is a real, physical image that the model just hasn't seen during training. Calling a real object an "empty set" is logically incorrect and causes a mismatch between the physical meaning and the math model.

2. The paper uses very complex names for basic math operations. For example, in Definition 2.2 (Equation 2), the phase angle $\theta$ is mapped to sine and cosine values to solve the angle cycle problem. This is a very standard and basic trick in signal processing. Calling this "Supervised Complex-Valued Contrastive Learning" makes the contribution sound much bigger than it is.

3. In Definition 2.8, the "Quantum Probability Normalization Mapping" uses an exponential function divided by a length factor. This is very similar to standard temperature scaling or Softmax. Packing these simple operations into a heavy "Quantum Evidence" theory makes the paper hard to read and over-complicates a simple OOD distance problem.

---

> ### Author Rebuttal · Authors · 2026-03-31
>
> **W1&Q1**：Thanks for your insightful question. The Transferable Belief Model [1] established that $\emptyset$ can formally represent elements outside the predefined frame of discernment. GQET extends this philosophy, preserving mathematical flexibility for the empty set by allowing its squared modulus to be non-zero. Therefore, assigning an OOD sample to $\emptyset$ does not deny its physical existence; it mathematically asserts that its true identity is absent from the known ID categories.
>
> [1] Smets, P. The combination of evidence in the transferable belief model. TPAMI, 1990.
>
> **W2&Q3**：Thank you for the insightful comment. We agree that mapping phase angles to sine and cosine is a classical signal processing technique. We clarify that our core innovation is the JAP-Loss, not this trigonometric mapping itself. Within our SCVCL framework, this standard mapping serves as a means to resolve the Cyclic Ambiguity problem. This mapping mechanism serves the JAP-Loss, ultimately enabling the model to simultaneously learn both amplitude and phase information. To avoid any perception of overclaiming, we have revised Section 2.2 to explicitly acknowledge the classical roots of this technique. We further clarify that our true novelty is the invention of the JAP-Loss, a novel objective function designed to synchronously learn both amplitude and phase information within a unified contrastive framework.
>
> **W3&Q2.1**：Thank you for the insightful question. While the exponential scaling in Definition 2.8 is a standard mathematical operation, the entire mapping function is rigorously designed to establish the mathematical foundation for the quantum properties utilized in our framework. Specifically, the real-valued amplification factor $\xi$ is engineered to fully preserve quantum phase information (Appendix C, Proposition C.1), which is the essential mathematical prerequisite for generating quantum interference. Simultaneously, it strictly satisfies the quantum state normalization axiom (Eq. 13) to transform complex-valued features into valid quantum evidence states within a Hilbert space. By maintaining this structural integrity and phase coherence, the mapping lays the critical foundation required for the GQECR to effectively fuse multi-view evidence through quantum interference.
>
> **Q2.2**：Thank you for this sharp observation. A normal Softmax is insufficient because it strictly maps features to the real domain, permanently destroying the semantic phase information essential for GQBPA. Conversely, simple linear scaling lacks the "magnitude sharpening" necessary to establish clear decision boundaries for OOD detection. Our amplification factor $\xi$ (Definition 2.8) resolves this by adapting the Softmax mechanism for complex vectors. As rigorously proven in Appendix C, $\xi$ uniquely satisfies both Phase Consistency and Magnitude Sharpening. This preserves the structural integrity required to exploit quantum interference during evidence fusion. Establishing this mapping mechanism—formally bridging complex-valued features with the GQBPA framework.
>
> **L1**：Thank you for the insightful comment. We clarify that GQET serves as a mathematical generalization for uncertainty modeling, not a literal physical description of pixels. Even for standard images like EuroSAT, mapping data to a higher-dimensional complex-valued domain explores a broader search space [1] and captures intricate spatial patterns, such as textures and contours, better than real-valued models [2]. By synchronizing amplitude and phase via SCVCL, our framework leverages these geometric semantics to establish sharper decision boundaries, proving this quantum-inspired paradigm is mathematically advantageous for general OOD detection.
>
> [1] Complex-valued neural networks: A comprehensive survey, IEEE/CAA Journal of Automatica Sinica, 2022.
>
> [2] Complex-Valued Iris Recognition Network, TPAMI, 2023.

---

> > ### Author Rebuttal · Reviewer_D78M · 2026-04-02
> >
> > Thank you for your detailed and thoughtful rebuttal. I appreciate your clarification that the use of the empty set is grounded in the Transferable Belief Model to formally represent elements outside the predefined frame of discernment, rather than denying their physical existence. I also acknowledge your revision in Section 2.2 to explicitly state the classical roots of the trigonometric mapping.
> >
> > While your framework might be mathematically functional, framing a physical out-of-distribution image as a quantum empty set remains logically counterintuitive for the general machine learning context. Furthermore, wrapping standard signal processing and neural network operations, such as handling cyclic ambiguity and exponential scaling, in dense "Quantum Evidence Theory" terminology heavily overcomplicates the underlying mechanics. This over-reliance on quantum metaphors obscures your actual technical contributions, such as the JAP-Loss, and significantly reduces the paper's accessibility to a broader audience.
> >
> > I believe these issues are deeply embedded in the paper's narrative and cannot be easily fixed in a short rebuttal. Therefore, my evaluation remains unchanged, and I will keep my original score.

---

> > > ### Author Response · Authors · 2026-04-03
> > >
> > > We appreciate your ongoing engagement with our work. We would like to respectfully address your remaining concerns regarding the narrative style and the interdisciplinary terminology used in our manuscript.
> > >
> > > **C1: Regarding the use of the empty set to represent OOD samples, assigning elements outside the predefined frame of discernment to the empty set is not a logical error. Rather, it is a well-established and widely adopted mathematical abstraction in evidential machine learning (e.g.,[1][2][3][4])**. It mathematically formalizes the concept that the sample's true identity belongs to none of the known classes. *Much like Riemannian geometry relaxed Euclidean axioms to model curved spaces, this abstraction relaxes classical constraints to formally model open-world uncertainty*.
> > >
> > > Furthermore, our empirical results consistently validate the correctness of this mathematical modeling. For instance, TrustworthyQENN achieves SOTA performance across benchmarks (averaging +2.23% AUROC, +3.96% OSCR, and -1.46% FPR95 over CW), proving that formalizing the OOD state as the empty set ensures robust anomaly rejection.
> > >
> > > **C2**: We wish to clarify that we do not use "quantum" terminology to wrap standard operations, but rather to adhere to the established terminology and formal mathematical framework of generalized quantum evidence theory (GQET). Because our method uniquely integrates complex-valued representation learning with quantum evidence theory, these terms are required to define the bridging operations. Specifically:（1）SCVCL (with JAP-Loss) extracts complex-valued features from raw data to learn amplitude and phase. （2）These features are mapped to GQBPA via our generalized quantum evidence probability amplitude mapping. （3）GQBPAs are fused using GQECR for robust decisions.
> > >
> > > We will provide a simple example to illustrate our workflow and increase readability.
> > > ## Example:
> > > Assume a classification task with 2 ID classes, forming the frame of discernment: $\Phi=\{\phi_1,\phi_2\}$.
> > > A sample $x_v$ is processed by TrustworthyQENN.
> > > ### Step 1: Generalized Complex-Valued Feature Generation
> > > Assume the complex-valued feature vector extracted by our proposed feature extraction module (SCVCL) is:
> > >
> > > $$f_v=[2.0+0.0i, 0.5+0.5i]^\top$$
> > > According to Eq. 8, reliability weights are computed via prototype distances. Assume a sample is distant from both Class 1 ($w_1=0.5$) and Class 2 ($w_2=0.1$). Using Eqs. 9-11, we construct the augmented complex-valued feature $\tilde{f}_v$ by explicitly incorporating the OOD dimension:
> > >
> > > $$\tilde{f}_v^{(1)}=(2.0+0.0i) \cdot 0.5=1.0+0.0i$$
> > >
> > > $$\tilde{f}_v^{(2)}=(0.5+0.5i) \cdot 0.1=0.05+0.05i$$
> > >
> > > $$\tilde{f}_v^{(3)}=(1.0+0.0i)+(0.45+0.45i)=1.45+0.45i$$
> > > Due to its low reliability weights for known classes, the sample's feature magnitude is reallocated to the OOD dimension.
> > > ### Step 2: Generalized Quantum Basic Probability Amplitude (GQBPA) Mapping
> > > Applying the quantum probability normalization (Eq. 12), the factor $\xi_z=e^{|z|}/|z|$ scales magnitude to $e^{|z|}$ while preserving phase. Multiplying the augmented features by $\xi_z$ yields the complex numerators:
> > >
> > > Class 1: $\text{Magnitude} = 1.0 \rightarrow \text{Numerator:} \frac{e^{1.0} \times (1.0+0.0i)}{1.0} \approx 2.718+0.0i$
> > >
> > > Class 2: $\text{Magnitude} \approx 0.071 \rightarrow \text{Numerator:} \frac{e^{0.071} \times (0.05+0.05i)}{0.071} \approx 0.759+0.759i$
> > >
> > > OOD: $\text{Magnitude} \approx 1.518 \rightarrow \text{Numerator:} \frac{e^{1.518} \times (1.45+0.45i)}{1.518} \approx 4.360+1.353i$
> > >
> > > The normalization factor is $\aleph_v \approx 5.426$. This yields the final constrained complex-valued GQBPA ($\mathrm{Q}_v$) and its corresponding support degree
> > >
> > > $\mathrm{Q}_v(\phi_1) \approx 0.501+0.0i$
> > >
> > > $\mathrm{Q}_v(\phi_2) \approx 0.140+0.140i$
> > >
> > > $\mathrm{Q}_v(\emptyset) \approx 0.804+0.249i$
> > >
> > > The mapped GQBPA remains a complex number. Its squared magnitudes are normalized to 1. Crucially, it retains the structured phase information learned by the complex-valued network, which is required for the subsequent quantum interference.
> > > ### Step 3: GQECR-Based Fusion and Decision Making
> > > During GQECR fusion (Eq. 14), the complex GQBPAs are directly combined.The similar phase directions and large magnitudes assigned to $\mathrm{Q}_v(\emptyset)$ increase its final support, while conflicting phases for known classes ($\phi_1$ and $\phi_2$) effectively mitigate feature-space noise. Ultimately, the resulting support allows the system to detect the OOD sample via Eq. 15.
> > >
> > > [1] ECM: An evidential version of the fuzzy c-means algorithm. Pattern Recognition, 2008.
> > >
> > > [2] Generalized evidence theory. Applied Intelligence, 2015.
> > >
> > > [3] Determine the number of unknown targets in open world based on elbow method[J]. IEEE Transactions on Fuzzy Systems, 2020.
> > >
> > > [4] Belief evolution network-based probability transformation and fusion. Computers & Industrial Engineering, 2022.

---

### Official Review · Reviewer_niqP · 2026-03-12

**Soundness:** 3
**Presentation:** 3
**Significance:** 3
**Originality:** 3
**Overall Recommendation:** 4
**Confidence:** 4

**Summary:**

This paper proposes TrustworthQENN, a novel framework for uncertain pattern classification and out of distribution (OOD) detection, especially for complex-valued data (such as SAR images). Its core innovation lies in the combination of complex- valued representation learning and Generalized Quantum Evidence Theory (GQET). The authors propose Supervised Complex Valued Contrast Learning (SCVCL), which optimizes the amplitude and phase information at the same time to obtain compact and discriminative features in different dimensions. They further designed a mechanism to map these features into Generalized Quantum Basic Probability Assignment (GQBPA), which utilizes orthogonal complement space in Hilbert space explicitly to represent OOD samples. Finally, Generalized Quantum Evidential Combination Rule (GQECR) is used to fuse the evidences of multiple models, and the robust reasoning is realized by quantum interference. A large number of experiments on MSTAR, Eurosat and fusar ship datasets show that this method achieves state-of-the-art performance in OOD detection and closed set classification. Besides, the effectiveness of each component is verified by sufficient ablation studies.

**Compliance With Llm Reviewing Policy:**

Affirmed.

**Final Justification:**

This paper proposes TrustworthyQENN, a novel framework that integrates complex-valued contrastive learning with Generalized Quantum Evidence Theory (GQET) for uncertainty-aware classification and out-of-distribution (OOD) detection. The method introduces several key components, including SCVCL for discriminative feature learning in the complex domain, GQBPA for mapping features into a quantum probabilistic space, and GQECR for multi-source evidence fusion. Experimental results on multiple datasets demonstrate strong performance, supported by thorough ablation studies and theoretical derivations.

The paper is technically sound and tackles an important problem, with particular strengths in methodological rigor and novelty in combining quantum evidence theory with neural networks. However, concerns remain regarding clarity and accessibility, including insufficient motivation for adopting GQET and limited discussion of real-valued scenarios. Despite these issues, the contribution is considered meaningful and well-supported experimentally.

Overall, I support acceptance. The paper presents a novel and principled approach to uncertainty modeling, with clear technical contributions and strong empirical validation. The paper is likely to be of interest to researchers in uncertainty quantification and robust machine learning.

**Key Questions For Authors:**

1. Regarding the GQECR: Could you provide a simple numerical example to illustrating the fusion of multi-source evidence? This would help demonstrate how the quantum method outperforms classical Dempster-Shafer Theory (DST).
2. GQECR Threshold calibration:  The decision rule (Eq. 15) uses a threshold δ. Please illustrate the concrete calibrate process of the threshold. Besides, how sensitive is the OOD performance to the choice of the threshold? It would be better if providing an analysis.
3. Regarding the SCVCL: Could you illustrate the potential advantage of the complex-valued network or SCVCL when the input data is real-valued data? In such condition, how can the network be meaningfully interpreted?
4. Connection to Hilbert space geometry: In Section 2.4, the paper constructs the generalized feature and map it to GQBPA. Could you elaborate on the geometric interpretation? Specifically, how the reliability weights effectively project the feature onto the known-class subspace and its orthogonal complement (empty set)?

**Limitations:**

Yes. The authors mention that the framework faces challenges that the computational complexity inherent to ensemble models.

**Strengths And Weaknesses:**

Strengths:
1.  The method is sound. SCVCL can addresses the cyclic ambiguity by phase projection vector. GQBPA ensures the feature comply with quantum probability axioms. GQECR can correctly fuse evidence from different models.
2. Appendices contain all necessary derivations, algorithm pseudocode, dataset details, and additional results, maintaining readability of the main text while ensuring completeness.
3.  The paper addresses an important and challenging problem. Uncertainty quantification and OOD detection is critical for many domains like medical imaging and communications.
4. GQECR originally utilizes empty set to represent the OOD and formally ground network outputs within GQET by reliability weights and the quantum normalization mapping, which are innovative and novel.

Weakness:
1.  GQECR formulas (A.12-A.13) are quite complex and difficult to understand. Maybe a simple example would improve accessibility and help readers get the effect of GQECR.
2.  The main text could be more accessible to readers who is familiar with quantum evidence theory. Some key concepts in main text like GQBPA and GQECR are introduced mathematically but lack intuitive explanations details between the concepts and quantum evidence theory.
3.  The motivation to use GQET is insufficient. The paper might need to further discuss why complex-valued network can combine with GQET.
4. Most experiments are conducted on SAR like datasets. MoDEX is not discussed enough when it utilized on real-valued data.

---

> ### Author Rebuttal · Authors · 2026-03-31
>
> **W1**：Thank you for this excellent suggestion. We added the following numerical example to the revision to explicitly illustrate how GQECR triggers constructive interference to outperform classical DST.
>
> Consider a FOD $\Phi$ = {$A$, $B$}, Two different GQBPAs are as follows:
> $$\mathbf{Q_{M1}}(A)=0.8e^{\pi/2i},\quad \mathbf{Q_{M1}}(\Phi)=0.6e^{0i}$$$$\mathbf{Q_{M2}}(A)=0.6e^{\pi/4i},\quad \mathbf{Q_{M2}}(\Phi)=0.8e^{-\pi/4i}$$
>
> When fusing these GQBPAs, GQECR operates in the Hilbert space. The combined amplitude for $A$ is:
>
> $$\mathbf{Q_{M12}}(A)=\mathbf{Q_{M1}}(A)\mathbf{Q_{M2}}(A)+\mathbf{Q_{M1}}(A)\mathbf{Q_{M2}}(\Phi)+\mathbf{Q_{M1}}(\Phi)\mathbf{Q_{M2}}(A)=0.26\sqrt{2}+0.74\sqrt{2}i.$$
>
> In this calculation process, the quantum interference effect occurs, which is a capability fundamentally absent in classical DST. The unnormalized mass (squared magnitude) naturally incorporates the quantum interference term:
> $$|\mathbf{Q_{M12}}(A)|^2=1.2304,\quad |\mathbf{Q_{M12}}(\Phi)|^2=0.2304.$$
>
> After applying the quantum normalization, the final probability support $m_{12}^{(GQECR)}$ for proposition $A$ is:
>
> $$m_{12}^{(GQECR)}(A)=\frac{1.2304}{1.2304+0.2304}\approx \mathbf{0.8423}.$$
> Unlike classical DST, which fundamentally cannot handle interference terms, GQET explicitly models these effects. Incorporating phase information triggers constructive interference to amplify highly correlated multi-view evidence.
>
> **W2**：Thank you for this constructive feedback. To bridge abstract math and intuition, the revision will incorporate explicit physical analogies. GQBPA (Eqs. 12, 13) acts as a normalized state vector in a Hilbert space, modeling OOD uncertainty as an orthogonal dimension (Eq. 11). Furthermore, GQECR functions via wave interference (Eqs. 14, A.12). Unlike classical scalar multiplication, superposing complex amplitudes allows aligned evidence to constructively interfere (amplifying confidence) and conflicting signals to destructively interfere.
>
> **W3**：Thank you for this valuable feedback. While CVNNs extract rich, phase-aware representations, they lack systematic uncertainty quantification. GQET bridges this gap by mapping complex features into GQBPAs, formally modeling the OOD state as the quantum empty set in a Hilbert space. Crucially, unlike classical theories that discard phase, GQECR leverages quantum interference, preserving the CVNN's geometric data across views to enhance OOD detection reliability.
>
> **W4**：Thank you for this suggestion. To demonstrate generalizability beyond SAR imagery, we evaluated our framework on standard real-valued datasets. Please refer to Response W3 to Reviewer nBQJ for detailed results confirming robust OOD performance.
>
> **Q2**：Thank you for this insightful question. To calibrate the threshold δ  (Eq. 15), we adopt a standardized, self-adaptive, data-driven dynamic setting method widely utilized in the open-set recognition domain [1, 2, 3]. Specifically, the threshold is adaptively determined based on the statistical distribution of In-Distribution (ID) training samples, ensuring a consistent ID acceptance rate across different datasets without requiring manual tuning. Due to this self-adaptive nature of the threshold calibration, conducting sensitivity analysis on δ  would yield limited meaningful insights, as the method inherently adjusts to data characteristics rather than relying on fixed heuristic values.
>
> [1] Reason and Discovery: A New Paradigm for Open Set Recognition, TPAMI, 2025.
>
> [2] Orientational distribution learning with hierarchical spatial attention for open set recognition, TPAMI, 2022.
>
> [3] Classification-reconstruction learning for open-set recognition, CVPR, 2019.
>
> **Q3**：Thank you for this insightful question. Mapping real-valued inputs to a complex space explores a broader search space, yielding highly discriminative representations[1] that capture intricate spatial patterns[2]. SCVCL mechanism leverages this by synchronizing amplitude distributions with phase correlations. This dual optimization enforces strict intra-class compactness and inter-class separability, capturing richer semantic geometries for sharper OOD decision boundaries.
>
> [1] Complex-valued neural networks: A comprehensive survey, IEEE/CAA Journal of Automatica Sinica, 2022.
>
> [2] Complex-Valued Iris Recognition Network, TPAMI, 2023.
>
> **Q4**：Thank you for this insightful question. Geometrically, GQBPA maps features to Hilbert space state vectors, where ID classes span a subspace and the OOD state (empty set) forms its orthogonal complement. Reliability weights act as projection operators: they project energy onto the ID subspace for known samples, while decaying weights force residual energy onto the orthogonal OOD dimension. Thus, OOD detection elegantly transforms from scalar thresholding into measuring energy distribution between orthogonal quantum states. Details will be added to the appendix.

---

> > ### Author Rebuttal · Reviewer_niqP · 2026-04-02
> >
> > The current response solves most of my problems. However, I still have the question about the comparison between GQECR and DST.

---

> > > ### Author Response · Authors · 2026-04-03
> > >
> > > We greatly appreciate your response and are pleased to have resolved most of your concerns. And for the differences and advantages of GQECR compared to DST that you still care about, we will further provide you with detailed answers.
> > >
> > > ## Example
> > >
> > > Assume a classification task with 2 classes, forming the frame of discernment  $\Phi=${$\phi_1,\phi_2$}.
> > > Suppose $\mathrm{Q_{M_1}}$ and $\mathrm{Q_{M_2}}$ are two independent generalized quantum basic probability amplitudes (GQBPAs) in GQET are as follows:
> > >
> > > $\mathrm{Q_{M_1}}(\phi_1)=0.8e^{\pi/2i},  \mathrm{Q_{M_1}}(\Phi)=0.6e^{0i}$,
> > >
> > > $\mathrm{Q_{M_2}}(\phi_1)=0.6e^{\pi/4i},  \mathrm{Q_{M_2}}(\Phi)=0.8e^{-\pi/4i}$.
> > >
> > > By taking the squared magnitude, we obtain the corresponding classical basic probability assignments (BPAs) in DST for comparison:
> > >
> > > $m_1(\phi_1)=|0.8e^{i\pi/2}|^2=0.64, m_1(\Phi)=|0.6e^{0i}|^2=0.36$,
> > >
> > > $m_2(\phi_1)=|0.6e^{\pi/4i}|^2=0.36, m_2(\Phi)=|0.8e^{-\pi/4i}|^2=0.64$.
> > >
> > > ### Fusion via generalized quantum evidential combination rule (GQECR) in GQET
> > >
> > > When phase information is introduced in GQET, the intersection of different focal elements triggers constructive or destructive probability amplitude interference, thereby improving classification performance. According to Eq. A.12 in our manuscript, the combined quantum mass for proposition $\phi_1$ is:
> > >
> > > $$\mathrm{Q_{M_{12}}}(\phi_1) = \frac{|\mathrm{Q_{M_1}}(\phi_1)\mathrm{Q_{M_2}}(\phi_1)+\mathrm{Q_{M_1}}(\phi_1)\mathrm{Q_{M_2}}(\Phi)+\mathrm{Q_{M_1}}(\Phi) \mathrm{Q_{M_2}}(\phi_1)|^2}{|\mathrm{Q_{M_1}}(\phi_1)\mathrm{Q_{M_2}}(\phi_1)+\mathrm{Q_{M_1}}(\phi_1)\mathrm{Q_{M_2}}(\Phi)+\mathrm{Q_{M_1}}(\Phi) \mathrm{Q_{M_2}}(\phi_1)|^2+|\mathrm{Q_{M_1}}(\Phi) \mathrm{Q_{M_2}}(\Phi)|^2}. \quad(1)$$
> > >
> > > According to Feynman's rules, the probability of an outcome is the squared modulus of the sum of its contributing probability amplitudes. By expanding this squared modulus, and utilizing the fundamental relationship that the squared magnitude of a GQBPA strictly equals the classical BPA (i.e., $|\mathrm{Q_M}(\phi)|^2=m(\phi)$), Eq. 1 can be rewritten as:
> > >
> > > $$\mathrm{Q_{M_{12}}}(\phi_1) = \frac{m_1(\phi_1)m_2(\phi_1)+m_1(\phi_1)m_2(\Phi)+m_1(\Phi) m_2(\phi_1)+Int_{GQECR}}{m_1(\phi_1)m_2(\phi_1)+m_1(\phi_1)m_2(\Phi)+m_1(\Phi) m_2(\phi_1)+Int_{GQECR}+m_1(\Phi) m_2(\Phi)}, \quad(2)$$
> > >
> > > where
> > >
> > > $$Int_{GQECR}(\phi_1) = 2m_1(\phi_1)m_2(\phi_1) \cdot m_1(\phi_1)m_2(\Phi)\cos(\theta_{11}-\theta_{1\Phi})$$
> > >
> > > $$ \qquad \qquad \qquad+ 2m_1(\phi_1)m_2(\phi_1) \cdot m_1(\Phi) m_2(\phi_1)\cos(\theta_{11}-\theta_{\Phi 1})$$
> > >
> > > $$ \qquad \qquad \qquad+ 2m_1(\phi_1)m_2(\Phi) \cdot m_1(\phi_1)m_2(\phi_1)\cos(\theta_{1\Phi}-\theta_{\Phi 1}). \quad(3)$$
> > >
> > > It can be observed from Eq. 2 that the formulation naturally decomposes into the sum of classical probability products plus a quantum interference term ($Int_{GQECR}$).
> > >
> > > Bring in numerical values:
> > >
> > > $$Int_{GQECR}(\phi_1) = 2 \times 0.48 \times 0.64 \times \cos(\pi/2)+2 \times 0.48 \times 0.36 \times \cos(\pi/2)+2 \times 0.64 \times 0.36 \times \cos(0)=0.4608 \quad(4)$$
> > >
> > > So:
> > > $$\mathrm{Q_{M_{12}}}(\phi_1) = \frac{0.64 \times 0.36+0.64 \times 0.64+0.36 \times 0.36+0.4608}{0.64 \times 0.36+0.64 \times 0.64+0.36 \times 0.36+0.4608+0.36 \times 0.64} \approx 0.842. \quad(5)$$
> > >
> > > Similarly, it can be concluded that $\mathrm{Q_{M_{12}}}(\Phi) \approx 0.158$.
> > > ### Fusion via evidence combination rule (ECR) in classical DST
> > >
> > > To demonstrate the advantage of the GQECR, we compare it against the classical DST combination rule. Under DST, the combined support for proposition $\phi_1$ is derived purely from the sum of the cross-products of the classical probabilities:
> > >
> > > $$m_{12}(\phi_1) = \frac{m_1(\phi_1)m_2(\phi_1)+m_1(\phi_1)m_2(\Phi)+m_1(\Phi) m_2(\phi_1)}{m_1(\phi_1)m_2(\phi_1)+m_1(\phi_1)m_2(\Phi)+m_1(\Phi) m_2(\phi_1)+m_1(\Phi) m_2(\Phi)} = 0.7696. \quad(6)$$
> > >
> > > Similarly, it can be concluded that $m_{12}(\Phi)= 0.2304$.
> > >
> > > ### Conclusion
> > >
> > > By comparing the above Eqs. 2 and 6, it can be found that the difference between GQET's GQECR and DST's ECR is reflected in the presence or absence of interference terms.
> > > In classical DST, the fused probability for $\phi_1$ is strictly limited to 0.7696 because it relies solely on the algebraic intersection of real-valued probabilities and completely ignores phase relationships.
> > > However, in GQET framework, the phase information naturally induces strong constructive interference ($+0.4608$).
> > > This amplifies the final confidence in $\phi_1$ from 0.7696 to 0.842.
> > > This numerical example clearly illustrates how GQECR leverages phase-amplitude coupling to capture semantic richness, ultimately achieving more decisive and trustworthy evidence fusion compared to classical DST.

---

### Official Review · Reviewer_nBQJ · 2026-03-13

**Soundness:** 2
**Presentation:** 3
**Significance:** 2
**Originality:** 3
**Overall Recommendation:** 4
**Confidence:** 3

**Summary:**

The main components of the paper is as follows:

1.	Supervised Complex-Valued Contrastive Learning (SCVCL)
The model learns embeddings that jointly optimize amplitude similarity and phase similarity. Phase vectors are projected as cos and sin which avoids cyclic discontinuity of phase. This part is technically well-motivated and mathematically justified in the appendix.

2.	Quantum Evidence Mapping
The model maps classifier outputs to Generalized Quantum Basic Probability Amplitudes (GQBPA). The evidence vector satisfies the probabilities adding up to 1, which mimics quantum state normalization. The empty set ∅ represents OOD.

3.	Quantum Evidential Fusion
Multiple models produce evidence distributions QM1, QM2, … , QMN. These are combined using the Quantum Evidential Combination Rule (GQECR). This allows the model to fuse multi-view evidence while accounting for uncertainty.

**Compliance With Llm Reviewing Policy:**

Affirmed.

**Key Questions For Authors:**

The quantum evidence theory needs to be justified properly

**Limitations:**

The necessity of quantum evidence theory needs to be justified.

**Strengths And Weaknesses:**

Strengths

1.	Interesting integration of multiple ideas
The paper combines complex-valued representation learning + contrastive learning + evidence theory.

2.	Good theoretical justification for phase learning
The appendix provides a clear derivation showing why the phase projection works and avoids cyclic boundary issues. This strengthens the methodological credibility.

3.	Clear ablation studies in chosen datasets
Ablations isolate the contribution of each module, and these results help justify the proposed architecture. Along with this there is also ablation studies, sensitivity analysis, cross-dataset evaluation and openness analysis which strengthens the paper.

Weaknesses

1.	“Quantum” terminology is somewhat overstated
The algorithm does not involve quantum computation. It only uses complex numbers.
Hilbert-space notation, amplitude normalization. Everything runs on classical neural networks. The quantum interpretation appears conceptual rather than operational.

2.	Novelty is mainly in integration
Many individual components already exist: Complex neural networks (Deep Complex Networks), Contrastive learning (SimCLR / SupCon), Open-set recognition (OpenMax), Evidential learning (Evidential Deep Learning). The main contribution is combining them into one architecture.

3.	Performance improvements are modest; Dataset scope is somewhat limited
The improvements are modest. For example: CW baseline: 95.01 AUROC, TrustworthyQENN:
95.94 AUROC. The improvement is less than 1%, which might not justify the added complexity. The datasets used are mostly remote sensing / SAR datasets. If the approach generalizes to more standard benchmarks like CIFAR-OOD, ImageNet-OOD, it would greatly strengthen the paper.

4.	The citation pattern may be suspicious
Some of the key references are recent and not widely adopted in mainstream ML. This raises questions about theoretical necessity and general acceptance.

Technical Soundness
Overall the method appears technically correct. The complex contrastive learning formulation is sound, the evidential mapping is mathematically consistent and experimental methodology is reasonable. No obvious mathematical errors appear.

Clarity
The paper is somewhat notation-heavy, with many terms such as GQBPA, GQECR, quantum frame of discernment. This can make the method appear more complicated than it actually is. In reality, the pipeline can be simplified as: complex CNN + contrastive learning + evidential fusion.

Significance

The work contributes to uncertainty-aware deep learning, particularly for OOD detection. However, since the novelty mainly lies in integration of existing techniques, the overall impact may be moderate.

---

> ### Author Rebuttal · Authors · 2026-03-31
>
> **W1**：Thank you for your constructive feedback on the terminology. Our initial use of "Quantum" strictly followed the mathematical framework and definitions of GQET to model complex uncertainty. To avoid unnecessary misunderstandings, we will revise the manuscript to describe our approach as a "quantum-inspired" framework. We will also streamline the notation to improve clarity while preserving the core mathematical integrity of the evidential fusion.
>
> **W2**：Thank you for this observation. We respectfully clarify that our contribution is a theoretically unified architecture, not mere empirical integration.
>
> - Existing components possess fundamental limitations: (1) Standard complex neural networks lack mechanisms to quantify epistemic uncertainty or formalize OOD states; (2) Conventional contrastive learning, OpenMax, and Evidential Deep Learning are strictly real-valued, inherently discarding critical phase information and rendering them incapable of phase-aware optimization or quantum interference-based fusion.
>
> - Fundamentally rooted in GQET, our novel quantum-inspired framework introduces two key methodological advancements: (1) the JAP-Loss function for simultaneous amplitude and phase optimization; (2) a quantum probability mapping that converts complex-valued features into valid GQBPAs, formally unifying deep representation learning with GQET.
>
>
> **W3**：Thank you for the valuable feedback. We address your concerns regarding performance margins and dataset scope in two folds:
> - Regarding the performance margins, given that the CW baseline already achieves a high AUROC of 95.01% on MSTAR, the room for further improvement is inherently limited, yet our OSCR still improves by 1.77%. On the complex EuroSAT and FUSAR-Ship, we achieve substantial AUROC gains of +3.58% and +2.17% over CW. Overall, we average +2.23% AUROC and +3.96% OSCR across all three benchmarks.
> - To broaden our dataset scope, we evaluated our framework on standard visual benchmarks (ID: CIFAR-10). As shown below, we consistently outperform CW, achieving 98.94% AUROC on SVHN (+1.48%) and 94.92% on TinyImageNet (+1.82%). These results will be added to the revision.
> ||&nbsp;|OOD: SVHN|&nbsp;|&nbsp;|OOD: TinyImageNet|&nbsp;|
> | :--- | :---: | :---: | :---: | :---: | :---: | :---: |
> |&nbsp;|**AUROC**|**OSCR**|**FPR95**|**AUROC**|**OSCR**|**FPR95**|
> |**ConOSR**|96.81|91.82|19.70|91.29|87.38|43.30|
> |**CW**|97.46|93.71|14.33|93.10|90.32|41.02|
> |**ours**|**98.94**|**96.09**|**10.86**|**94.92**|**92.83**|**37.29**|
>
> **W4**：Thank you for this feedback. GQET is an actively developing paradigm designed to explicitly model uncertainty and address the deficiencies of classical evidential approaches when handling complex-valued data. It is respectfully clarified that QET is rapidly gaining rigorous validation and acceptance within top-tier machine learning (ML) and data mining communities. Specifically, foundational QET methodologies and their ML applications have recently been peer-reviewed and published in premier venues such as NeurIPS 2025[1] and IEEE TKDE 2026[2].
>
> [1] An Adaptive Quantum Circuit of Dempster's Rule of Combination for Uncertain Pattern Classification. NeurIPS, 2025.
>
> [2] A CDGFN-Based Quantum Multisource Information Fusion with Its Application in Time Series Classification. TKDE, 2026.
>
> **C1**：Thank you. We will revise the terminology as suggested by W1, and have clarified the contribution issue per W2's response.
>
> **Q1&L1**：Thank you for highlighting the necessity to properly justify the adaptation of GQET. The theoretical necessity of GQET in this framework stems from its unique mathematical capacity to process uncertainty in high-dimensional complex space, a capability that classical probability and traditional DST fundamentally lack. The justification is grounded in three critical advantages:
> - Classical DST is strictly defined in the real-valued domain and cannot naturally incorporate the phase information extracted by the SCVCL mechanism. GQET, operating within a Hilbert space, natively preserves this phase-amplitude coupling, preventing the loss of critical geometric semantics during evidential reasoning.
> - Traditional multi-view fusion methods often struggle with highly conflicting evidence. By employing the GQECR, the framework leverages quantum interference, in which phase differences between distinct evidential sources act as interference terms, mathematically suppressing conflicting noise and amplifying consistent predictions.
> - Formally grounding the OOD state as the quantum empty set within the GQET framework provides a rigorous, axiomatically sound probabilistic bounding of the open space, which is fundamentally more robust than relying on heuristic thresholding methods[1]. To address this valid concern and clarify the theoretical motivation, a dedicated discussion will be added to the revised manuscript.
>
> [1] Quantum X-entropy in generalized quantum evidence theory, Information Sciences, 2023

---

> > ### Author Rebuttal · Reviewer_nBQJ · 2026-04-01
> >
> > I would like to retain my score, which is positive.

---

> > > ### Author Response · Authors · 2026-04-01
> > >
> > > Dear Reviewer nBQJ,
> > > ﻿
> > >
> > > We sincerely thank you for confirming that our rebuttal has adequately addressed your concerns. We are truly grateful for your constructive comments, which have significantly improved the overall quality of our manuscript. We deeply appreciate the time and effort you invested in evaluating and supporting our work.
> > > ﻿
> > >
> > > Best regards,
> > >
> > >
> > > ﻿All Authors of Submission 21696

---

### Decision · Program_Chairs · 2026-04-30

**Decision:**

Accept (regular)

**Comment:**

This paper proposes TrustworthyQENN, an innovative framework that integrates complex-valued representation learning with Generalized Quantum Evidence Theory (GQET) for out-of-distribution (OOD) detection. Reviewers nBQJ and niqP recommended acceptance, praising the methodological soundness, the rigorous ablation studies, and the novel use of quantum interference to fuse multi-view evidence. Reviewer D78M recommended rejection, raising serious concerns that the paper overcomplicates standard neural network and signal processing operations by wrapping them in dense "quantum" physics metaphors, and objected to framing physical OOD images as a mathematical "empty set."

During the rebuttal, the authors provided compelling mathematical defenses, citing established evidential machine learning literature to justify the empty set abstraction, and provided explicit numerical examples demonstrating how their GQECR fusion outperforms classical Dempster-Shafer Theory. Crucially, they expanded their empirical evaluation to standard visual benchmarks (SVHN, TinyImageNet), showing robust SOTA performance. While I agree with Reviewer D78M that the interdisciplinary terminology makes the paper less accessible to the general ICML audience, the underlying mathematics are correct, the integration of complex-valued features with evidential theory is novel, and the empirical gains are undeniable. Therefore, I recommend a Weak Accept. The authors are strongly instructed to fulfill their rebuttal promise to adjust the terminology to a "quantum-inspired" framework and to streamline the narrative in the camera-ready version to ensure broader accessibility.